# Joint quantile regression in vector-valued RKHSs

**Maxime Sangnier      Olivier Fercoq      Florence d'Alché-Buc**
LTCI, CNRS, Télécom ParisTech
Université Paris-Saclay
75013, Paris, France
{maxime.sangnier, olivier.fercoq, florence.dalche}
@telecom-paristech.fr

## Abstract

Addressing the will to give a more complete picture than an average relationship provided by standard regression, a novel framework for estimating and predicting simultaneously several conditional quantiles is introduced. The proposed methodology leverages kernel-based multi-task learning to curb the embarrassing phenomenon of quantile crossing, with a one-step estimation procedure and no postprocessing. Moreover, this framework comes along with theoretical guarantees and an efficient coordinate descent learning algorithm. Numerical experiments on benchmark and real datasets highlight the enhancements of our approach regarding the prediction error, the crossing occurrences and the training time.

## 1   Introduction

Given a couple $(X, Y)$ of random variables, where $Y$ takes scalar values, a common aim in statistics and machine learning is to estimate the conditional expectation $\mathbb{E}\left[Y \mid X = x\right]$ as a function of $x$. In the previous setting, called regression, one assumes that the main information in $Y$ is a scalar value corrupted by a centered noise. However, in some applications such as medicine, economics, social sciences and ecology, a more complete picture than an average relationship is required to deepen the analysis. Expectiles and quantiles are different quantities able to achieve this goal.

This paper deals with this last setting, called (conditional) quantile regression. This topic has been championed by Koenker and Bassett [16] as the minimization of the pinball loss (see [15] for an extensive presentation) and brought to the attention of the machine learning community by Takeuchi et al. [26]. Ever since then, several studies have built upon this framework and the most recent ones include regressing a single quantile of a random vector [12]. On the contrary, we are interested in estimating and predicting simultaneously several quantiles of a scalar-valued random variable $Y|X$ (see Figure 1), thus called *joint quantile regression*. For this purpose, we focus on non-parametric hypotheses from a vector-valued Reproducing Kernel Hilbert Space (RKHS).

Since quantiles of a distribution are closely related, joint quantile regression is subsumed under the field of multi-task learning [3]. As a consequence, vector-valued kernel methods are appropriate for such a task. They have already been used for various applications, such as structured classification [10] and prediction [7], manifold regularization [21, 6] and functional regression [14]. Quantile regression is a new opportunity for vector-valued RKHSs to perform in a multi-task problem, along with a loss that is different from the $\ell_2$ cost predominantly used in the previous references.

In addition, such a framework offers a novel way to curb the phenomenon of quantile curve crossing, while preserving the so called *quantile property* (which may not be true for current approaches). This one guarantees that the ratio of observations lying below a predicted quantile is close to the quantile level of interest.

In a nutshell, the contributions of this work are (following the outline of the paper): **i)** a novel methodology for joint quantile regression, that is based on vector-valued RKHSs; **ii)** enhanced predictions thanks to a multi-task approach along with limited appearance of crossing curves; **iii)** theoretical guarantees regarding the generalization of the model; **iv)** an efficient coordinate descent algorithm, that is able to handle the intercept of the model in a manner that is simple and different from Sequential Minimal Optimization (SMO). Besides these novelties, the enhancements of the proposed method and the efficiency of our learning algorithm are supported by numerical experiments on benchmark and real datasets.

## 2 Problem definition

### 2.1 Quantile regression

Let $\mathcal{Y} \subset \mathbb{R}$ be a compact set, $\mathcal{X}$ be an arbitrary input space and $(X, Y) \in \mathcal{X} \times \mathcal{Y}$ a pair of random variables following an unknown joint distribution. For a given probability $\tau \in (0, 1)$, the conditional $\tau$-quantile of $(X, Y)$ is the function $\mu_\tau \colon \mathcal{X} \to \mathbb{R}$ such that $\mu_\tau(\mathbf{x}) = \inf\{\mu \in \mathbb{R} : \mathbb{P}(Y \leq \mu \mid X = \mathbf{x}) \geq \tau\}$. Thus, given a training set $\{(\boldsymbol{x_i}, y_i)\}_{i=1}^n \in (\mathcal{X} \times \mathcal{Y})^n$, the quantile regression problem aims at estimating this conditional $\tau$-quantile function $\mu_\tau$. Following Koenker [15], this can be achieved by minimization of the *pinball loss*: $\ell_\tau(r) = \max(\tau r, (\tau - 1)r)$, where $r \in \mathbb{R}$ is a residual. Using such a loss first arose from the observation that the location parameter $\mu$ that minimizes the $\ell_1$-loss $\sum_{i=1}^n |y_i - \mu|$ is an estimator of the unconditional median [16].

Now focusing on the estimation of a conditional quantile, one can show that the target function $\mu_\tau$ is a minimizer of the $\tau$-quantile risk $R_\tau(h) = \mathbb{E}[\ell_\tau(Y - h(X))]$ [17]. However, since the joint probability of $(X, Y)$ is unknown but we are provided with an independent and identically distributed (*iid*) sample of observations $\{(\boldsymbol{x_i}, y_i)\}_{i=1}^n$, we resort to minimizing the empirical risk: $R_\tau^{\mathrm{emp}}(h) = \frac{1}{n} \sum_{i=1}^n \ell_\tau(y_i - h(\mathbf{x}_i))$, within a class $\mathcal{H} \subset (\mathbb{R})^{\mathcal{X}}$ of functions, calibrated in order to overcome the shift from the true risk to the empirical one. In particular, when $\mathcal{H}$ has the form: $\mathcal{H} = \{h = f + b : b \in \mathbb{R}, f \in (\mathbb{R})^{\mathcal{X}}, \psi(f) \leq c\}$, with $\psi \colon (\mathbb{R})^{\mathcal{X}} \to \mathbb{R}$ being a convex function and $c > 0$ a constant, Takeuchi et al. [26] proved that (similarly to the unconditional case) the *quantile property* is satisfied: for any estimator $\hat{h}$, obtained by minimizing $R_\tau^{\mathrm{emp}}$ in $\mathcal{H}$, the ratio of observations lying below $\hat{h}$ (*i.e.* $y_i < \hat{h}(\mathbf{x}_i)$) equals $\tau$ to a small error (the ration of observations exactly equal to $\hat{h}(\mathbf{x}_i)$). Moreover, under some regularity assumptions, this quantity converges to $\tau$ when the sample grows. Note that these properties are true since the intercept $b$ is unconstrained.

### 2.2 Multiple quantile regression

In many real problems (such as medical reference charts), one is not only interested by estimating a single quantile curve but a few of them. Thus, denoting $\mathbb{N}_p$ the range of integers between 1 and $p$, for several quantile levels $\tau_j$ ($j \in \mathbb{N}_p$) and functions $h_j \in \mathcal{H}$, the empirical loss to be minimized can bi written as the following separable function: $R_{\boldsymbol{\tau}}^{\mathrm{emp}}(h_1, \ldots, h_p) = \frac{1}{n} \sum_{i=1}^n \sum_{j=1}^p \ell_{\tau_j}(y_i - h_j(\mathbf{x}_i))$, where $\boldsymbol{\tau}$ denotes the $p$ dimensional vector of quantile levels.

A nice feature of multiple quantile regression is thus to extract slices of the conditional distribution of $Y|X$. However, when quantiles are estimated independently, an embarrassing phenomenon often appears: quantile functions cross, thus violating the basic principle that the cumulative distribution function should be monotonically non-decreasing. We refer to that pitfall as the *crossing problem*.

In this paper, we propose to prevent curve crossing by considering the problem of multiple quantile regression as a vector-valued regression problem where outputs are not independent. An interesting feature of our method is to preserve the quantile property while most other approaches lose it when struggling to the crossing problem.

### 2.3 Related work

Going beyond linear and spline-based models, quantile regression in RKHSs has been introduced a decade ago [26, 17]. In [26], the authors proposed to minimize the pinball loss in a scalar-valued RKHS and to add hard constraints on the training points in order to prevent the crossing problem. Our work can be legitimately seen as an extension of [26] to multiple quantile regression using

a vector-valued RKHS and structural constraints against curve crossing thanks to an appropriate matrix-valued kernel.

Another related work is [27], which first introduced the idea of multi-task learning for quantile regression. In [27], linear quantile curves are estimated jointly with a common feature subspace shared across the tasks, based on multi-task feature learning [3]. In addition, the authors showed that for such linear regressors, a common representation shared across infinitely many tasks can be computed, thus estimating simultaneously conditional quantiles for all possible quantile levels. Both previous approaches will be considered in the numerical experiments.

Quantile regression has been investigated from many perspectives, including different losses leading to an approximate quantile property ($\epsilon$-insensitive [25], re-weighted least squares [22]) along with models and estimation procedures to curb the crossing problem: location-scale model with a multi-step strategy [13], tensor product spline surface [22], non-negative valued kernels [18], hard non-crossing constraints [26, 28, 5], inversion and monotonization of a conditional distribution estimation [9] and rearrangement of quantile estimations [8], to cite only a few references. Let us remark that some solutions such as non-crossing constraints [26] lose theoretically the quantile property because of constraining the intercept.

In comparison to the literature, we propose a novel methodology, based on vector-valued RKHSs, with a one-step estimation, no post-processing, and keeping the quantile property while dealing with curve crossing. We also provide an efficient learning algorithm and theoretical guarantees.

## 3 Vector-valued RKHS for joint quantile regression

### 3.1 Joint estimation

Given a vector $\boldsymbol{\tau} \in (0,1)^p$ of quantile levels, multiple quantile regression is now considered as a joint estimation in $(\mathbb{R}^p)^{\mathcal{X}}$ of the target function $\mathbf{x} \in \mathcal{X} \mapsto (\mu_{\tau_1}(\mathbf{x}), \ldots, \mu_{\tau_p}(\mathbf{x})) \in \mathbb{R}^p$ of conditional quantiles. Thus, let now $\psi$ be a convex regularizer on $(\mathbb{R}^p)^{\mathcal{X}}$ and $\mathcal{H} = \{h = f + \boldsymbol{b} : \boldsymbol{b} \in \mathbb{R}^p, f \in (\mathbb{R}^p)^{\mathcal{X}}, \psi(f) \leq c\}$ be the hypothesis set. Similarly to previously, joint quantile regression aims at minimizing $R_{\boldsymbol{\tau}}^{\mathrm{emp}}(h) = \frac{1}{n}\sum_{i=1}^n \ell_{\boldsymbol{\tau}}(y_i \mathbb{1} - h(\mathbf{x}_i))$, where $\mathbb{1}$ stands for the all-ones vector, $\ell_{\boldsymbol{\tau}}(\boldsymbol{r}) = \sum_{j=1}^p \ell_{\tau_j}(r_j)$ and $h$ is in $\mathcal{H}$, which is to be appropriately chosen in order to estimate the $p$ conditional quantiles while enhancing predictions and avoiding curve crossing. It is worthwhile remarking that, independently of the choice of $\psi$, the quantile property is still verified for a vector-valued estimator since the loss is separable and the intercept is unconstrained. Similarly, the vector-valued function whose components are the conditional $\tau_j$-quantiles is still a minimizer of the $\boldsymbol{\tau}$-quantile risk $R_{\boldsymbol{\tau}}(h) = \mathbb{E}\left[\ell_{\boldsymbol{\tau}}(Y\mathbb{1} - h(X))\right]$.

In this context, the constraint $\psi$ does not necessarily apply independently on each coordinate function $h_j$ but can impose dependency between them. The theory of vector-valued RKHS seems especially well suited for this purpose when considering $\psi$ as the norm associated to it. In this situation, the choice of the kernel does not only influence the nature of the hypotheses (linear, non-linear, universal approximators) but also the way the estimation procedure is regularized. In particular, the kernel critically operates on the output space by encoding structural constraints on the outputs.

### 3.2 Matrix-valued kernel

Let us denote $\cdot^\top$ the transpose operator and $\mathcal{L}(\mathbb{R}^p)$ the set of linear and bounded operators from $\mathbb{R}^p$ to itself. In our (finite) case, $\mathcal{L}(\mathbb{R}^p)$ comes down to the set of $p \times p$ real-valued matrices. A matrix-valued kernel is a function $K : \mathcal{X} \times \mathcal{X} \to \mathcal{L}(\mathbb{R}^p)$, that is symmetric and positive [20]: $\forall (\mathbf{x}, \mathbf{x}') \in \mathcal{X} \times \mathcal{X}, K(\mathbf{x}, \mathbf{x}') = K(\mathbf{x}', \mathbf{x})^\top$ and $\forall m \in \mathbb{N}, \forall \{(\boldsymbol{\alpha}_i, \boldsymbol{\beta}_i)\}_{1 \leq i \leq m} \in (\mathcal{X} \times \mathbb{R}^p)^m$, $\sum_{1 \leq i,j \leq m} \langle \boldsymbol{\beta}_i \mid K(\boldsymbol{\alpha}_i, \boldsymbol{\alpha}_j)\boldsymbol{\beta}_j \rangle_{\ell_2} \geq 0$.

Let $K$ be such a kernel and for any $\mathbf{x} \in \mathcal{X}$, let $K_\mathbf{x} : \boldsymbol{y} \in \mathbb{R}^p \mapsto K_\mathbf{x}\boldsymbol{y} \in (\mathbb{R}^p)^{\mathcal{X}}$ be the linear operator such that: $\forall \mathbf{x}' \in \mathcal{X}, (K_\mathbf{x}\boldsymbol{y})(\mathbf{x}') = K(\mathbf{x}', \mathbf{x})\boldsymbol{y}$. There exists a unique Hilbert space of functions $\mathcal{K}_K \subset (\mathbb{R}^p)^{\mathcal{X}}$ (with an inner product and a norm respectively denoted $\langle \cdot \mid \cdot \rangle_{\mathcal{K}}$ and $\|\cdot\|_{\mathcal{K}}$), called the RKHS associated to $K$, such that $\forall \mathbf{x} \in \mathcal{X}$ [20]: $K_\mathbf{x}$ spans the space $\mathcal{K}_K$ ($\forall \boldsymbol{y} \in \mathbb{R}^p : K_\mathbf{x}\boldsymbol{y} \in \mathcal{K}$), $K_\mathbf{x}$ is bounded for the uniform norm ($\sup_{\boldsymbol{y} \in \mathbb{R}^p} \|K_\mathbf{x}\boldsymbol{y}\|_{\mathcal{K}} < \infty$) and $\forall f \in \mathcal{K} : f(\mathbf{x}) = K_\mathbf{x}^* f$ (reproducing property), where $\cdot^*$ is the adjoint operator.

From now on, we assume that we are provided with a matrix-valued kernel $K$ and we limit the hypothesis space to: $\mathcal{H} = \{f + \boldsymbol{b} : \boldsymbol{b} \in \mathbb{R}^p, f \in \mathcal{K}_K, \|f\|_{\mathcal{K}} \leq c\}$ (*i.e.* $\psi = \|\cdot\|_{\mathcal{K}}$). Though several candidates are available [1], we focus on one of the simplest and most efficiently computable kernels, called *decomposable kernel*: $K : (\mathbf{x}, \mathbf{x}') \mapsto k(\mathbf{x}, \mathbf{x}')\boldsymbol{B}$, where $k : \mathcal{X} \times \mathcal{X} \to \mathbb{R}$ is a scalar-valued kernel and $\boldsymbol{B}$ is a $p \times p$ symmetric Positive Semi-Definite (PSD) matrix. In this particular case, the matrix $\boldsymbol{B}$ encodes the relationship between the components $f_j$ and thus, the link between the different conditional quantile estimators. A rational choice is to consider $\boldsymbol{B} = \left(\exp(-\gamma(\tau_i - \tau_j)^2)\right)_{1 \leq i,j \leq p}$. To explain it, let us consider two extreme cases (see also Figure 1).

First, when $\gamma = 0$, $\boldsymbol{B}$ is the all-ones matrix. Since $\mathcal{K}_K$ is the closure of the space span $\{K_{\mathbf{x}}\boldsymbol{y} : (\mathbf{x}, \boldsymbol{y}) \in \mathcal{X} \times \mathbb{R}^p\}$, any $f \in \mathcal{K}_K$ has all its components equal. Consequently, the quantile estimators $h_j = f_j + b_j$ are parallel (and non-crossing) curves. In this case, the regressor is said *homoscedastic*. Second, when $\gamma \to +\infty$, then $\boldsymbol{B} \to \boldsymbol{I}$ (identity matrix). In this situation, it is easy to show that the components of $f \in \mathcal{K}_K$ are independent from each other and that $\|f\|_{\mathcal{K}}^2 = \sum_{j=1}^p \|f_j\|_{\mathcal{K}'}^2$ (where $\|\cdot\|_{\mathcal{K}'}$ is the norm coming with the RKHS associated to $k$) is separable. Thus, each quantile function is learned independently from the others. Regressors are said *heteroscedastic*. It appears clearly that between these two extreme cases, there is a room for learning a non-homoscedastic and non-crossing quantile regressor (while preserving the quantile property).

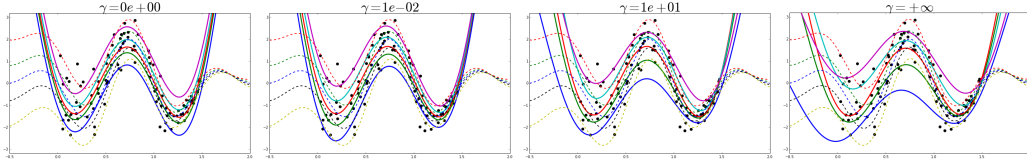

Figure 1: Estimated (plain lines) and true (dashed lines) conditional quantiles of $Y|X$ (synthetic dataset) from homoscedastic regressors ($\gamma = 0$) to heteroscedastic ones ($\gamma \to +\infty$).

## 4 Theoretical analysis

This section is intended to give a few theoretical insights about the expected behavior of our hypotheses. Here, we do assume working in an RKHS but not specifically with a decomposable kernel. First, we aim at providing a uniform generalization bound. For this purpose, let $\mathcal{F} = \{f \in \mathcal{K}_K, \|f\|_{\mathcal{K}} \leq c\}$, $\mathrm{tr}(\cdot)$ be the trace operator, $((X_i, Y_i))_{1 \leq i \leq n} \in (\mathcal{X} \times \mathcal{Y})^n$ be an *iid* sample and denote $\hat{R}_n(h) = \frac{1}{n} \sum_{i=1}^n \ell_{\boldsymbol{\tau}}(Y_i \mathbb{1} - h(X_i))$, the random variable associated to the empirical risk of a hypothesis $h$.

**Theorem 4.1** (Generalization). *Let $a \in \mathbb{R}_+$ such that $\sup_{y \in \mathcal{Y}} |y| \leq a$, $\boldsymbol{b} \in \mathcal{Y}^p$ and $\mathcal{H} = \{f + \boldsymbol{b} : f \in \mathcal{F}\}$ be the class of hypotheses. Moreover, assume that there exists $\kappa \geq 0$ such that: $\sup_{\mathbf{x} \in \mathcal{X}} \mathrm{tr}(K(\mathbf{x}, \mathbf{x})) \leq \kappa$. Then with probability at least $1 - \delta$ (for $\delta \in (0, 1]$):*

$$\forall h \in \mathcal{H}, \quad R(h) \leq \hat{R}_n(h) + 2\sqrt{2}c\sqrt{\frac{p\kappa}{n}} + (2pa + c\sqrt{p\kappa})\sqrt{\frac{\log(1/\delta)}{2n}}.$$

*Sketch of proof (full derivation in Appendix A.1).* We start with a concentration inequality for scalar-valued functions [4] and we use a vector-contraction property [19]. The bound on the Rademacher complexity of [24, Theorem 3.1] concludes the proof. □

The uniform bound in Theorem 4.1 states that, with high probability, all the hypotheses of interest have a true risk which is less that an empirical risk to an additive bias in $O(1/\sqrt{n})$. Let us remark that it makes use of the output dimension $p$. However, there exist non-uniform generalization bounds for operator-valued kernel-based hypotheses, which do not depend on the output dimension [14], being thus well-suited for infinite-dimensional output spaces. Yet those results, only hold for optimal solutions $\hat{h}$ of the learning problem, which we never obtain in practice.

As a second theoretical insight, Theorem 4.2 gives a bound on the quantile property, which is similar to the one provided in [26] for scalar-valued functions. This one states that $\mathbb{E}\left[\mathbb{P}\left(Y \leq h_j(X) \mid X\right)\right]$ does not deviate to much from $\tau_j$.

**Theorem 4.2** (Quantile deviation)**.** *Let us consider that the assumptions of Theorem 4.1 hold. Moreover, let $\epsilon > 0$ be an artificial margin, $\Gamma_\epsilon^+ : r \in \mathbb{R} \mapsto \mathrm{proj}_{[0,1]}\left(1 - \frac{r}{\epsilon}\right)$ and $\Gamma_\epsilon^- : r \in \mathbb{R} \mapsto \mathrm{proj}_{[0,1]}\left(-\frac{r}{\epsilon}\right)$, two ramp functions, $j \in \mathbb{N}_p$ and $\delta \in (0,1]$. Then with probability at least $1 - \delta$:*

$$\forall h \in \mathcal{H}, \quad \frac{1}{n}\sum_{i=1}^n \Gamma_\epsilon^-(Y_i - h_j(X_i)) - \Delta \leq \mathbb{E}\left[\mathbb{P}\left(Y \leq h_j(X) \mid X\right)\right] \leq \underbrace{\frac{1}{n}\sum_{i=1}^n \Gamma_\epsilon^+(Y_i - h_j(X_i))}_{\approx \tau_j} + \Delta,$$

*where $\Delta = \frac{2c}{\epsilon}\sqrt{\frac{\kappa}{n}} + \sqrt{\frac{\log(2/\delta)}{2n}}$.*

*Sketch of proof (full derivation in Appendix A.2).* The proof is similar to the one of Theorem 4.1, when remarking that $\Gamma_\epsilon^+$ and $\Gamma_\epsilon^-$ are $1/\epsilon$-Lipschitz continuous. $\qquad\square$

# 5   Optimization algorithm

In order to finalize the M-estimation of a non-parametric function, we need a way to jointly solve the optimization problem of interest and compute the estimator. For ridge regression in vector-valued RKHSs, representer theorems enable to reformulate the hypothesis $f$ and to derive algorithms based on matrix inversion [20, 6] or Sylvester equation [10]. Since the optimization problem we are tackling is quite different, those methods do not apply. Yet, deriving a dual optimization problem makes it possible to hit the mark.

Quantile estimation, as presented in this paper, comes down to minimizing a regularized empirical risk, defined by the pinball loss $\ell_{\boldsymbol{\tau}}$. Since this loss function is non-differentiable, we introduce slack variables $\boldsymbol{\xi}$ and $\boldsymbol{\xi}^*$ to get the following primal formulation. We also consider a regularization parameter $C$ to be tuned:

$$\underset{\substack{f\in\mathcal{K}_K, \boldsymbol{b}\in\mathbb{R}^p, \\ \boldsymbol{\xi},\boldsymbol{\xi}^*\in(\mathbb{R}^p)^n}}{\text{minimize}} \quad \frac{1}{2}\|f\|_{\mathcal{K}}^2 + C\sum_{i=1}^n\left(\langle\boldsymbol{\tau}\mid\boldsymbol{\xi}_i\rangle_{\ell_2} + \langle\mathbb{1}-\boldsymbol{\tau}\mid\boldsymbol{\xi}_i^*\rangle_{\ell_2}\right) \quad \text{s.t.} \left\{\begin{array}{l}\forall i\in\mathbb{N}_n: \boldsymbol{\xi}_i\succcurlyeq 0, \boldsymbol{\xi}_i^*\succcurlyeq 0, \\ y_i - f(\mathbf{x}_i) - \boldsymbol{b} = \boldsymbol{\xi}_i - \boldsymbol{\xi}_i^*,\end{array}\right. \quad (1)$$

where $\succcurlyeq$ is a pointwise inequality. A dual formulation of Problem (1) is (see Appendix B):

$$\underset{\boldsymbol{\alpha}\in(\mathbb{R}^p)^n}{\text{minimize}} \quad \frac{1}{2}\sum_{i,j=1}^n\langle\boldsymbol{\alpha}_i\mid K(\mathbf{x}_i,\mathbf{x}_j)\boldsymbol{\alpha}_j\rangle_{\ell_2} - \sum_{i=1}^n y_i\langle\boldsymbol{\alpha}_i\mid\mathbb{1}\rangle_{\ell_2} \quad \text{s.t.}\left\{\begin{array}{l}\sum_{i=1}^n\boldsymbol{\alpha}_i = 0_{\mathbb{R}^p}, \ \forall i\in\mathbb{N}_n: \\ C(\boldsymbol{\tau}-\mathbb{1})\preccurlyeq\boldsymbol{\alpha}_i\preccurlyeq C\boldsymbol{\tau},\end{array}\right. \quad (2)$$

where the linear constraints come from considering an intercept $\boldsymbol{b}$. The Karush-Kuhn-Tucker (KKT) conditions of Problem (1) indicate that a minimizer $\hat{f}$ of (1) can be recovered from a solution $\hat{\boldsymbol{\alpha}}$ of (2) with the formula $\hat{f} = \sum_{i=1}^n K_{\mathbf{x}_i}\hat{\boldsymbol{\alpha}}_i$. Moreover, $\hat{\boldsymbol{b}}$ can also be obtained thanks to KKT conditions. However, as we deal with a numerical approximate solution $\boldsymbol{\alpha}$, in practice $\boldsymbol{b}$ is computed by solving Problem (1) with $f$ fixed. This boils down to taking $b_j$ as the $\tau_j$-quantile of $(y_i - \hat{f}_j(\mathbf{x}_i))_{1\leq i\leq n}$.

Problem (2) is a common quadratic program that can be solved with off-the-shelf solvers. However, since we are essentially interested in decomposable kernels $K(\cdot,\cdot) = k(\cdot,\cdot)\boldsymbol{B}$, it appears that the quadratic part of the objective function would be defined by the $np \times np$ matrix $\boldsymbol{K}\otimes\boldsymbol{B}$, where $\otimes$ is the Kronecker product and $\boldsymbol{K} = (k(\mathbf{x}_i,\mathbf{x}_j))_{1\leq i,j\leq n}$. Storing this matrix explicitly is likely to be time and memory expensive. In order to improve the estimation procedure, ad hoc algorithms can be derived. For instance, regression with a decomposable kernel boils down to solving a Sylvester equation (which can be done efficiently) [10] and vector-valued Support Vector Machine (SVM) without intercept can be learned with a coordinate descent algorithm [21]. However, these methods can not be used in our setting since the loss function is different and considering the intercept is necessary for the quantile property. Yet, coordinate descent could theoretically be extended in an SMO technique, able to handle the linear constraints introduced by the intercept. However, SMO works usually with a single linear constraint and needs heuristics to run efficiently, which are quite difficult to find (even though an implementation exists for two linear constraints [25]).

Therefore, for the sake of efficiency, we propose to use a Primal-Dual Coordinate Descent (PDCD) technique, recently introduced in [11]. This algorithm (which is proved to converge) is able to deal with the linear constraints coming from the intercept and is thus utterly workable for the problem at hand. Moreover, PDCD has been proved favorably competitive with SMO for SVMs.

Table 1: Empirical pinball loss and crossing loss $\times 100$ (the less, the better). Bullets (resp. circles) indicate statistically significant (resp non-significant) differences. The proposed method is JQR.

| Data set | - | \multicolumn{4}{Pinball loss} | | | | - | \multicolumn{4}{Crossing loss} | | | |
|---|---|---|---|---|---|---|---|---|---|---|
| | - | IND. | IND. (NC) | MTFL | JQR | - | IND. | IND. (NC) | MTFL | JQR |
| caution | - | $102.6 \pm 17.3$ | $103.2 \pm 17.2$ | $102.9 \pm 19.0$ | ∘∘∘ $\mathbf{102.6 \pm 19.0}$ | - | $0.53 \pm 0.67$ | $0.31 \pm 0.70$ | $0.69 \pm 0.54$ | •∘• $\mathbf{0.09 \pm 0.14}$ |
| ftcollinssnow | - | $151.1 \pm 8.2$ | $\mathbf{150.8 \pm 8.0}$ | $152.4 \pm 8.9$ | ∘∘∘ $153.7 \pm 12.1$ | - | $\mathbf{0.00 \pm 0.00}$ | $0.00 \pm 0.00$ | $0.00 \pm 0.00$ | ∘∘∘ $0.00 \pm 0.00$ |
| highway | - | $102.9 \pm 39.1$ | $102.8 \pm 38.9$ | $\mathbf{102.0 \pm 34.5}$ | ∘∘∘ $103.7 \pm 35.7$ | - | $9.08 \pm 7.38$ | $9.00 \pm 7.39$ | $\mathbf{3.48 \pm 4.49}$ | ∘∘• $8.81 \pm 7.46$ |
| heights | - | $128.2 \pm 2.4$ | $128.2 \pm 2.4$ | $128.6 \pm 2.2$ | ∘∘• $\mathbf{127.9 \pm 1.8}$ | - | $0.04 \pm 0.05$ | $0.04 \pm 0.05$ | $0.07 \pm 0.14$ | ••• $\mathbf{0.00 \pm 0.00}$ |
| sniffer | - | $44.8 \pm 6.7$ | $\mathbf{44.6 \pm 6.8}$ | $46.9 \pm 7.6$ | ∘∘• $45.2 \pm 6.9$ | - | $1.01 \pm 0.75$ | $0.52 \pm 0.48$ | $1.23 \pm 0.77$ | ••• $\mathbf{0.15 \pm 0.22}$ |
| snowgeese | - | $68.4 \pm 35.3$ | $\mathbf{68.4 \pm 35.3}$ | $75.3 \pm 38.2$ | ∘∘∘ $76.0 \pm 31.5$ | - | $3.24 \pm 5.10$ | $2.60 \pm 4.28$ | $8.93 \pm 19.52$ | ••∘ $\mathbf{0.94 \pm 3.46}$ |
| ufc | - | $81.8 \pm 4.6$ | $81.6 \pm 4.6$ | $84.9 \pm 4.7$ | ••• $\mathbf{80.6 \pm 4.1}$ | - | $0.24 \pm 0.22$ | $0.27 \pm 0.42$ | $0.82 \pm 1.47$ | ••• $\mathbf{0.05 \pm 0.15}$ |
| birthwt | - | $139.0 \pm 9.9$ | $\mathbf{139.0 \pm 9.9}$ | $142.6 \pm 11.6$ | ∘∘∘ $139.8 \pm 11.7$ | - | $0.00 \pm 0.00$ | $0.00 \pm 0.00$ | $0.31 \pm 0.88$ | ••• $\mathbf{0.00 \pm 0.00}$ |
| crabs | - | $12.3 \pm 1.0$ | $12.3 \pm 1.0$ | $12.6 \pm 1.0$ | ••• $\mathbf{11.9 \pm 0.9}$ | - | $0.46 \pm 0.33$ | $0.35 \pm 0.24$ | $0.30 \pm 0.22$ | ••• $\mathbf{0.06 \pm 0.20}$ |
| GAGurine | - | $62.6 \pm 8.2$ | $62.6 \pm 8.2$ | $64.5 \pm 7.5$ | ∘∘• $\mathbf{62.6 \pm 8.1}$ | - | $0.05 \pm 0.08$ | $0.04 \pm 0.07$ | $0.05 \pm 0.09$ | ∘∘∘ $\mathbf{0.03 \pm 0.08}$ |
| geyser | - | $110.2 \pm 7.8$ | $110.1 \pm 7.8$ | $\mathbf{109.4 \pm 7.1}$ | ∘∘∘ $111.3 \pm 8.2$ | - | $0.87 \pm 1.60$ | $0.92 \pm 2.02$ | $0.80 \pm 1.18$ | ∘∘∘ $\mathbf{0.72 \pm 1.51}$ |
| gilgais | - | $47.4 \pm 4.4$ | $47.2 \pm 4.4$ | $49.9 \pm 3.6$ | ∘∘• $\mathbf{46.9 \pm 4.6}$ | - | $1.23 \pm 0.96$ | $0.95 \pm 0.85$ | $\mathbf{0.71 \pm 0.96}$ | ∘∘∘ $0.81 \pm 0.43$ |
| topo | - | $71.1 \pm 13.0$ | $70.1 \pm 13.7$ | $73.1 \pm 11.8$ | ∘∘∘ $\mathbf{69.6 \pm 13.4}$ | - | $2.72 \pm 3.26$ | $1.52 \pm 2.47$ | $2.75 \pm 2.93$ | •∘• $\mathbf{1.14 \pm 2.02}$ |
| BostonHousing | - | $48.5 \pm 5.0$ | $48.5 \pm 5.0$ | $49.7 \pm 4.7$ | ••• $\mathbf{47.4 \pm 4.7}$ | - | $0.64 \pm 0.32$ | $\mathbf{0.48 \pm 0.27}$ | $1.11 \pm 0.33$ | ∘•• $0.58 \pm 0.34$ |
| CobarOre | - | $0.5 \pm 0.5$ | $\mathbf{0.5 \pm 0.5}$ | $5.0 \pm 4.9$ | ••• $0.6 \pm 0.5$ | - | $0.10 \pm 0.13$ | $0.10 \pm 0.13$ | $0.29 \pm 0.35$ | ••• $\mathbf{0.02 \pm 0.05}$ |
| engel | - | $61.3 \pm 18.3$ | $61.2 \pm 19.0$ | $\mathbf{58.7 \pm 17.9}$ | ∘∘• $64.4 \pm 23.2$ | - | $1.50 \pm 4.94$ | $1.25 \pm 4.53$ | $1.65 \pm 5.97$ | ∘∘∘ $\mathbf{0.06 \pm 0.14}$ |
| mcycle | - | $89.2 \pm 8.5$ | $88.9 \pm 8.4$ | $102.0 \pm 11.7$ | ••• $\mathbf{84.3 \pm 10.3}$ | - | $2.10 \pm 1.83$ | $0.92 \pm 1.25$ | $1.13 \pm 1.10$ | ••• $\mathbf{0.14 \pm 0.37}$ |
| BigMac2003 | - | $71.0 \pm 21.0$ | $70.9 \pm 21.1$ | $68.7 \pm 18.1$ | ••∘ $\mathbf{67.6 \pm 20.9}$ | - | $2.50 \pm 2.12$ | $1.87 \pm 1.68$ | $\mathbf{0.73 \pm 0.92}$ | ∘∘∘ $1.55 \pm 1.75$ |
| UN3 | - | $99.5 \pm 7.0$ | $99.4 \pm 7.0$ | $101.8 \pm 7.1$ | ∘∘• $\mathbf{98.8 \pm 7.6}$ | - | $1.06 \pm 0.85$ | $0.85 \pm 0.70$ | $0.65 \pm 0.62$ | ••• $\mathbf{0.09 \pm 0.31}$ |
| cpus | - | $20.0 \pm 13.7$ | $19.9 \pm 13.6$ | $23.8 \pm 16.0$ | ∘∘• $\mathbf{19.7 \pm 13.7}$ | - | $1.29 \pm 1.13$ | $1.17 \pm 1.15$ | $0.46 \pm 0.28$ | ••• $\mathbf{0.09 \pm 0.13}$ |

PDCD is described in Algorithm 1, where, for $\boldsymbol{\alpha} = (\boldsymbol{\alpha}_i)_{1 \leq i \leq n} \in (\mathbb{R}^p)^n$, $\boldsymbol{\alpha}^j \in \mathbb{R}^n$ denotes its $j^{\text{th}}$ *row* vector and $\alpha_i^j$ its $i^{\text{th}}$ component, $\mathrm{diag}$ is the operator mapping a vector to a diagonal matrix and $\mathrm{proj}_{\mathbb{1}}$ and $\mathrm{proj}_{[C(\tau_l-1), C\tau_l]}$ are respectively the projectors onto the vector $\mathbb{1}$ and the compact set $[C(\tau_l - 1), C\tau_l]$. PDCD uses dual variables $\boldsymbol{\theta} \in (\mathbb{R}^p)^n$ (which are updated during the descent) and has two sets of parameters $\boldsymbol{\nu} \in (\mathbb{R}^p)^n$ and $\boldsymbol{\mu} \in (\mathbb{R}^p)^n$, that verify $(\forall (i,l) \in \mathbb{N}_n \times \mathbb{N}_p)$: $\mu_i^l < \frac{1}{(K(\mathbf{x}_i, \mathbf{x}_i))_{l,l} + \nu_i^l}$. In practice, we kept the same parameters as in [11]: $\nu_i^l = 10(K(\mathbf{x}_i, \mathbf{x}_i))_{l,l}$ and $\mu_i^l$ equal to 0.95 times the bound. Moreover, as it is standard for coordinate descent methods, our implementation uses efficient updates for the computation of both $\sum_{j=1}^n K(\mathbf{x}_i, \mathbf{x}_j) \boldsymbol{\alpha}_j$ and $\overline{\boldsymbol{\theta}}^l$.

## 6 Numerical experiments

Two sets of experiments are presented, respectively aimed at assessing the ability of our methodology to predict quantiles and at comparing an implementation of Algorithm 1 with an off-the-shelf solver and an augmented Lagrangian scheme. Following the previous sections, a decomposable kernel $K(\mathbf{x}, \mathbf{x}') = k(\mathbf{x}, \mathbf{x}')\boldsymbol{B}$ is used, where $\boldsymbol{B} = (\exp(-\gamma(\tau_i - \tau_j)^2))_{1 \leq i, j \leq p}$ and $k(\mathbf{x}, \mathbf{x}') = \exp(-\|\mathbf{x} - \mathbf{x}'\|_{\ell_2}^2 / 2\sigma^2)$, with $\sigma$ being the 0.7-quantile of the pairwise distances of the training data $\{\mathbf{x}_i\}_{1 \leq i \leq n}$. Quantile levels of interest are $\boldsymbol{\tau} = (0.1, 0.3, 0.5, 0.7, 0.9)$.

### 6.1 Quantile regression

Quantile regression is assessed with two criteria: the pinball loss $\frac{1}{n}\sum_{i=1}^n \ell_{\boldsymbol{\tau}}(y_i \mathbb{1} - h(\mathbf{x}_i))$ is the one minimized to build the proposed estimator and the crossing loss $\sum_{j=1}^{p-1} \left[ \frac{1}{n} \sum_{i=1}^n \max(0, h_{j+1}(\mathbf{x}_i) - h_j(\mathbf{x}_i)) \right]$, assuming that $\tau_j > \tau_{j+1}$, quantifies how far $h_j$ goes below $h_{j+1}$, while $h_j$ is expected to stay always above $h_{j+1}$. More experiments are in Appendix D.1.

This study focuses on three non-parametric models based on the RKHS theory. Other linear and spline-based models have been dismissed since Takeuchi et al. [26] have already provided a comparison of these ones with kernel methods. First, we considered an independent estimation of quantile regressors (IND.), which boils down to setting $\boldsymbol{B} = \boldsymbol{I}$ (this approach could be set up without vector-valued RKHSs but with scalar-valued kernels only). Second, hard non-crossing constraints on the training data have been imposed (IND. (NC)), as proposed in [26]. Third, the proposed joint estimator (JQR) uses the Gaussian matrix $\boldsymbol{B}$ presented above.

Quantile regression with multi-task feature learning (MTFL), as proposed in [27], is also included. For a fair comparison, each point is mapped with $\psi(\mathbf{x}) = (k(\mathbf{x}, \mathbf{x}_1), \ldots, k(\mathbf{x}, \mathbf{x}_n))$ and the estimator $h(\mathbf{x}) = \boldsymbol{W}^\top \psi(\mathbf{x}) + \boldsymbol{b}$ ($\boldsymbol{W} \in \mathbb{R}^{n \times p}$) is learned jointly with the PSD matrix $\boldsymbol{D} \in \mathbb{R}^{n \times n}$ of the

**Algorithm 1** Primal-Dual Coordinate Descent.

Initialize $\boldsymbol{\alpha}_i, \boldsymbol{\theta}_i \in \mathbb{R}^p$ ($\forall i \in \mathbb{N}_n$).
**repeat**
    Choose $(i, l) \in \mathbb{N}_n \times \mathbb{N}_p$ uniformly at random.
    Set $\overline{\boldsymbol{\theta}}^l \leftarrow \text{proj}_{\mathbb{1}} \left( \boldsymbol{\theta}^l + \text{diag}(\boldsymbol{\nu}^l)\boldsymbol{\alpha}^l \right)$.
    Set $d_i^l \leftarrow \sum_{j=1}^n (K(\mathbf{x}_i, \mathbf{x}_j)\boldsymbol{\alpha}_j)^l - y_i + 2\overline{\theta}_i^l - \theta_i^l$.
    Set $\overline{\alpha}_i^l \leftarrow \text{proj}_{[C(\tau_l-1), C\tau_l]} \left( \alpha_i^l - \mu_i^l d_i^l \right)$.
    Update coordinate $(i, l)$: $\alpha_i^l \leftarrow \overline{\alpha}_i^l, \theta_i^l \leftarrow \overline{\theta}_i^l$,
    and keep other coordinates unchanged.
**until** duality gap (1)-(2) is small enough

Table 2: CPU time (s) for training a model.

| Size | QP | AUG. LAG. | PDCD |
|---|---|---|---|
| 250 | **8.73** $\pm$ 0.34 | 261.11 $\pm$ 46.69 | 18.69 $\pm$ 3.54 |
| 500 | 75.53 $\pm$ 2.98 | 865.86 $\pm$ 92.26 | **61.30** $\pm$ 7.05 |
| 1000 | 621.60 $\pm$ 30.37 | – | **266.50** $\pm$ 41.16 |
| 2000 | 3416.55 $\pm$ 104.41 | – | **958.93** $\pm$ 107.80 |

regularizer $\psi(h) = \text{tr}(\boldsymbol{W}^\top \boldsymbol{D}^{-1} \boldsymbol{W})$. This comes down to alternating our approach (with $\boldsymbol{B} = \boldsymbol{I}$ and $k(\cdot, \cdot) = \langle \cdot \mid \boldsymbol{D}\cdot \rangle_{\ell_2}$) and the update $\boldsymbol{D} \leftarrow (\boldsymbol{W}\boldsymbol{W}^\top)^{1/2} / \text{tr}((\boldsymbol{W}\boldsymbol{W}^\top)^{1/2})$.

To present an honorable comparison of these four methods, we did not choose datasets for the benefit of our method but considered the ones used in [26]. These 20 datasets (whose names are indicated in Table 1) come from the UCI repository and three R packages: quantreg, alr3 and MASS. The sample sizes vary from 38 (CobarOre) to 1375 (heights) and the numbers of explanatory variables vary from 1 (5 sets) to 12 (BostonHousing). The datasets were standardized coordinate-wise to have zero mean and unit variance. Results are given in Table 1 thanks to the mean and the standard deviation of the test losses recorded on 20 random splits train-test with ratio 0.7-0.3. The best result of each line is boldfaced and the bullets indicate the significant differences of each competitor from JQR (based on a Wilcoxon signed-rank test with significance level 0.05).

The parameter $C$ is chosen by cross-validation (minimizing the pinball loss) inside a logarithmic grid $(10^{-5}, 10^{-4}, \ldots, 10^5)$ for all methods and datasets. For our approach (JQR), the parameter $\gamma$ is chosen in the same grid as $C$ with extra candidates $0$ and $+\infty$. Finally, for a balanced comparison, the dual optimization problems corresponding to each approach are solved with CVXOPT [2].

Regarding the pinball loss, joint quantile regression compares favorably to independent and hard non-crossing constraint estimations for 12 vs 8 datasets (5 vs 1 significantly different). These results bear out the assumption concerning the relationship between conditional quantiles and the usefulness of multiple-output methods for quantile regression. Prediction is also enhanced compared to MTFL for 15 vs 5 datasets (11 vs 1 significantly different).

The crossing loss clearly shows that joint regression enables to weaken the crossing problem, in comparison to independent estimation and hard non-crossing constraints (18 vs 1 favorable datasets and 9 vs 0 significantly different). Results are similar compared to MTFL (16 vs 3, 12 vs 1). Note that for IND. (NC), the crossing loss is null on the training data by construction but not necessarily on the test data. In addition, let us remark that model selection (and particularly for $\gamma$, which tunes the trade-off between hetero and homoscedastic regressors) has been performed based on the pinball loss only. It seems that, in a way, the pinball loss embraces the crossing loss as a subcriterion.

## 6.2 Learning algorithms

This section is aimed at comparing three implementations of algorithms for estimating joint quantile regressors (solving Problem 2), following their running (CPU) time. First, the off-the-shelf solver (based on an interior-point method) included in CVXOPT [2] (QP) is applied to Problem (2) turned into a standard form of linearly constrained quadratic program. Second, an augmented Lagrangian scheme (AUG. LAG) is used in order to get rid of the linear constraints and to make it possible to use a coordinate descent approach (detailed procedure in Appendix C). In this scheme, the inner solver is Algorithm 1 when the intercept is dismissed, which boils down to be the algorithm proposed in [23]. The last approach (PDCD) is Algorithm 1.

We use a synthetic dataset (the same as in Figure 1), for which $X \in [0, 1.5]$. The target $Y$ is computed as a sine curve at 1 Hz modulated by a sine envelope at 1/3 Hz and mean 1. Moreover, this pattern is distorted with a random Gaussian noise with mean 0 and a linearly decreasing standard deviation from 1.2 at $X = 0$ to 0.2 at $X = 1.5$. Parameters for the models are: $(C, \gamma) = (10^2, 10^{-2})$.

To compare the implementations of the three algorithms, we first run QP, with a relative tolerance set to $10^{-2}$, and store the optimal objective value. Then, the two other methods (AUG. LAG and PDCD) are launched and stopped when they pass the objective value reached by QP (optimal objective values are reported in Appendix D.2). Table 2 gives the mean and standard deviation of the CPU time required by each method for 10 random datasets and several sample sizes. Some statistics are missing because AUG. LAG. ran out of time.

As expected, it appears that for a not too tight tolerance and big datasets, implementation of Algorithm 1 outperforms the two other competitors. Let us remark that QP is also more expensive in memory than the coordinate-based algorithms like ours. Moreover, training time may seem high in comparison to usual SVMs. However, let us first remind that we jointly learn $p$ regressors. Thus, a fair comparison should be done with an SVM applied to an $np \times np$ matrix, instead of $n \times n$. In addition, there is no sample sparsity in quantile regression, which does speed up SVM training.

Last but not least, in order to illustrate the use of our algorithm, we have run it on two 2000-point datasets from economics and medicine: the U.S. 2000 Census data, consisting of annual salary and 9 related features on workers, and the 2014 National Center for Health Statistics' data, regarding girl birth weight and 16 statistics on parents.[1] Parameters $(C, \gamma)$ have been set to $(1, 100)$ and $(0.1, 1)$ respectively for the Census and NCHS datasets (determined by cross-validation). Figure 2 depicts 9 estimated conditional quantiles of the salary with respect to the education (17 levels from no schooling completed to doctorate degree) and of the birth weight (in grams) vs mother's pre-pregnancy weight (in pounds). As expected, the Census data reveal an increasing and heteroscedastic trend while new-born's weight does not seem correlated to mother's weight.

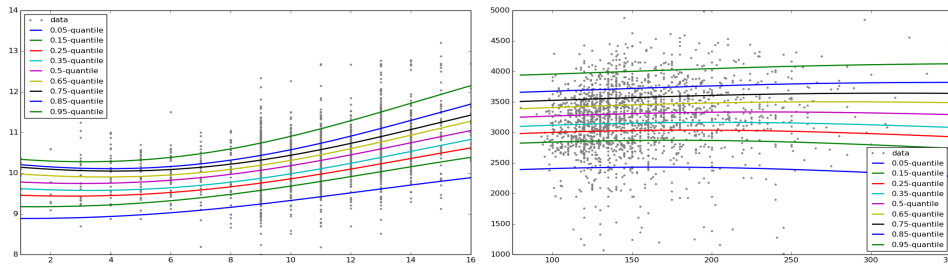

Figure 2: Estimated conditional quantiles for the Census (left, salary vs education) and the NCHS data (right, birth weight vs mother's pre-pregnancy weight).

## 7 Conclusion

This paper introduces a novel framework for joint quantile regression, which is based on vector-valued RKHSs. It comes along with theoretical guarantees and an efficient learning algorithm. Moreover, this methodology, which keeps the quantile property, enjoys few curve crossing and enhanced performances compared to independent estimations and hard non-crossing constraints.

To go forward, let us remark that this framework benefits from all the tools now associated with vector-valued RKHSs, such as manifold learning for the semi-supervised setting, multiple kernel learning for measuring feature importance and random Fourier features for very large scale applications. Moreover, extensions of our methodology to multivariate output variables are to be investigated, given that it requires to choose among the various definitions of multivariate quantiles.

**Acknowledgments**

This work was supported by the industrial chair "Machine Learning for Big Data".

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
