[Supplementary Material · supplementary.pdf]

# Supplementary: Joint quantile regression in vector-valued RKHSs

**Maxime Sangnier**  **Olivier Fercoq**  **Florence d'Alché-Buc**
LTCI, CNRS, Télécom ParisTech
Université Paris-Saclay
75013, Paris, France
{maxime.sangnier, olivier.fercoq, florence.dalche}
@telecom-paristech.fr

## A  Detailed proofs

### A.1  Generalization

This section describes a proof of the generalization bound given in the corpus of the paper. The result is based on a concentration inequality à la Bartlett and Mendelson [1] for vector-valued functions [4], along with a bound on the Rademacher complexity of operator-valued kernel based hypothesis sets [7]. Before stating these two critical properties, let us remind the definition of the Rademacher complexity, used to quantify the complexity of a class of functions.

**Definition A.1** (Rademacher complexity [4]). *Let $(X_i)_{1 \le i \le n} \in \mathcal{X}^n$ be an independent and identically distributed (*iid*) sample of random variables and $(\epsilon_{i,j})_{\substack{1 \le i \le n \\ 1 \le j \le p}} \in \{-1, 1\}^{n \times p}$ be $n \times p$ independent Rademacher variables (i.e. uniformly distributed on $\{-1, 1\}$). Let now $\mathcal{F} \subset (\mathbb{R}^p)^{\mathcal{X}}$ be a class of functions from $\mathcal{X}$ to $\mathbb{R}^p$. The Rademacher complexity (or average) of the class $\mathcal{F}$ is defined as:*

$$\mathcal{R}_n(\mathcal{F}) = \mathbb{E}\left[ \sup_{f \in \mathcal{F}} \frac{1}{n} \sum_{\substack{1 \le i \le n \\ 1 \le j \le p}} \epsilon_{i,j} f_j(X_i) \right],$$

*where the expectation is computed jointly on $(X_i)_{1 \le i \le n}$ and $(\epsilon_{i,j})_{\substack{1 \le i \le n \\ 1 \le j \le p}}$.*

**Proposition A.1** (Concentration for Lipschitz hypotheses). *Let $X \in \mathcal{X}$ and $(X_i)_{1 \le i \le n} \in \mathcal{X}^n$ be* iid *random variables, $\mathcal{F} \subset (\mathbb{R}^p)^{\mathcal{X}}$ a class of functions. Let $\phi \colon \mathbb{R}^p \to [a, b]$ $(a, b \in \mathbb{R})$ be a Lipschitz continuous mapping with Lipschitz constant $L_\phi$:*

$$\forall (\boldsymbol{z}, \boldsymbol{z}') \in \mathbb{R}^p \colon |\phi(\boldsymbol{z}) - \phi(\boldsymbol{z}')| \le L_\phi \|\boldsymbol{z} - \boldsymbol{z}'\|_{\ell_2}.$$

*Let $\delta \in (0, 1]$, then with probability at least $1 - \delta$:*

$$\sup_{f \in \mathcal{F}} \left( \mathbb{E}\left[\phi(f(X))\right] - \frac{1}{n} \sum_{i=1}^{n} \phi(f(X_i)) \right) \le 2\sqrt{2} L_\phi \mathcal{R}_n(\mathcal{F}) + (b - a)\sqrt{\frac{\log(1/\delta)}{2n}}.$$

*Proof.* Thanks to the assumptions above, with probability at least $1 - \delta$, we have [1, 3]:

$$\sup_{f \in \mathcal{F}} \left( \mathbb{E}\left[\phi(f(X))\right] - \frac{1}{n} \sum_{i=1}^{n} \phi(f(X_i)) \right) \le 2\mathcal{R}_n(\Phi) + (b - a)\sqrt{\frac{\log(1/\delta)}{2n}},$$

where $\Phi = \{\phi \circ f : f \in \mathcal{F}\}$. Then, using [5, Corollary 1], we obtain:

$$\mathcal{R}_n(\Phi) \le \sqrt{2} L_\phi \mathcal{R}_n(\mathcal{F}).$$

Gathering both equations concludes the proof. ∎

**Proposition A.2** (Bound on the Rademacher average [7, Theorem 3.1]). *Assume that there exists $\kappa \in \mathbb{R}_+$ such that: $\sup_{\mathbf{x} \in \mathcal{X}} \operatorname{tr}(K(\mathbf{x}, \mathbf{x})) \leq \kappa$ and let $\mathcal{F} = \{f \in \mathcal{K}_K, \|f\|_{\mathcal{K}} \leq c\}$ for a given $c \in \mathbb{R}_+$. Then:*

$$\mathcal{R}_n(\mathcal{F}) \leq c\sqrt{\frac{\kappa}{n}}.$$

**Theorem A.3** (Generalization). *Let $\boldsymbol{\tau} \in (0,1)^p$, $((X_i, Y_i))_{1 \leq i \leq n} \in (\mathcal{X} \times \mathcal{Y})^n$ be iid random variables (independent from $(X, Y)$), $a \in \mathbb{R}_+$ such that $\sup_{y \in \mathcal{Y}} |y| \leq a$ and $\boldsymbol{b} \in \mathcal{Y}^p$. Let $\mathcal{H} = \{f + \boldsymbol{b} \colon f \in \mathcal{K}_K / \|f\|_{\mathcal{K}} \leq c\}$, for a given $c \in \mathbb{R}_+$, be the class of hypotheses. Moreover, assume that there exists $\kappa \in \mathbb{R}_+$ such that: $\sup_{\mathbf{x} \in \mathcal{X}} \operatorname{tr}(K(\mathbf{x}, \mathbf{x})) \leq \kappa$ and, for a hypothesis $h$, let us denote*

$$\hat{R}_n(h) = \frac{1}{n} \sum_{i=1}^n \ell_{\boldsymbol{\tau}}(Y_i \mathbb{1} - h(X_i)),$$

*the random variable associated to the empirical risk. Let $\delta \in (0,1]$, then with probability at least $1 - \delta$:*

$$\forall h \in \mathcal{H} \colon R(h) \leq \hat{R}_n(h) + 2\sqrt{2}c\sqrt{\frac{p\kappa}{n}} + (2pa + c\sqrt{p\kappa})\sqrt{\frac{\log(1/\delta)}{2n}}.$$

*Proof.* Let $\mathcal{F} = \{f \in \mathcal{K}_K, \|f\|_{\mathcal{K}} \leq c\}$. The proof begins with the following lemma.

**Lemma A.4.** *Under the assumptions of Theorem A.3: $\forall (f, \mathbf{x}) \in \mathcal{F} \times \mathcal{X}, \|f(\mathbf{x})\|_{\ell_2} \leq c\sqrt{\kappa}$.*

*Proof of Lemma A.4.*

$$\forall (f, \mathbf{x}, \boldsymbol{y}) \in \mathcal{F} \times \mathcal{X} \times \mathbb{R}^p,$$
$$\begin{aligned} \langle f(\mathbf{x}) \mid \boldsymbol{y} \rangle_{\ell_2} &= \langle K_{\mathbf{x}} \boldsymbol{y} \mid f \rangle_{\mathcal{K}} \\ &\leq \|f\|_{\mathcal{K}} \|K_{\mathbf{x}} \boldsymbol{y}\|_{\mathcal{K}} \\ &\leq c\sqrt{\langle K_{\mathbf{x}} \boldsymbol{y} \mid K_{\mathbf{x}} \boldsymbol{y} \rangle_{\mathcal{K}}} \\ &= c\sqrt{\langle \boldsymbol{y} \mid K(\mathbf{x}, \mathbf{x})\boldsymbol{y} \rangle_{\ell_2}}. \end{aligned}$$

From the properties of operator-valued kernels, we know that $K(\mathbf{x}, \mathbf{x})$ is symmetric positive semi-definite. Thus, denoting $(\lambda_j)_{1 \leq j \leq p}$ its (non-negative) eigenvalues and $(e_j)_{1 \leq j \leq p}$ the corresponding orthonormal basis, we obtain when $\|\boldsymbol{y}\|_{\ell_2} \leq 1$:

$$\begin{aligned} \langle \boldsymbol{y} \mid K(\mathbf{x}, \mathbf{x})\boldsymbol{y} \rangle_{\ell_2} &= \sum_{i,j} \langle \boldsymbol{y} \mid e_i \rangle_{\ell_2} \langle \boldsymbol{y} \mid e_j \rangle_{\ell_2} \langle e_i \mid K(\mathbf{x}, \mathbf{x})e_j \rangle_{\ell_2} \\ &= \sum_j \langle \boldsymbol{y} \mid e_j \rangle_{\ell_2}^2 \lambda_j \\ &\leq \sum_j \lambda_j \qquad\qquad\qquad \text{(since } \|\boldsymbol{y}\|_{\ell_2} \leq 1\text{)} \\ &= \operatorname{tr}(K(\mathbf{x}, \mathbf{x})). \end{aligned}$$

Thus: $\forall (f, \mathbf{x}) \in \mathcal{F} \times \mathcal{X}, \|f(\mathbf{x})\|_{\ell_2} = \sup_{\|\boldsymbol{y}\|_{\ell_2} \leq 1} \langle f(\mathbf{x}) \mid \boldsymbol{y} \rangle_{\ell_2} \leq c\sqrt{\kappa}.$ □

In order to apply Proposition A.1, let us observe that the loss function $\ell_\tau$ is $\sqrt{p}$-Lipschitz:

$$
\forall (\boldsymbol{r}, \boldsymbol{r}') \in \mathbb{R}^p,
$$
$$
\ell_\tau(\boldsymbol{r}) = \ell_\tau(\boldsymbol{r} - \boldsymbol{r}' + \boldsymbol{r}')
$$
$$
= \sum_{j=1}^{p} \max\left(\tau_j(r_j - r'_j + r'_j), (\tau_j - 1)(r_j - r'_j + r'_j)\right)
$$
$$
= \sum_{j=1}^{p} \left\{ \begin{array}{ll} \tau_j(r_j - r'_j + r'_j) & \text{if } r_j - r'_j + r'_j \geq 0 \\ (\tau_j - 1)(r_j - r'_j + r'_j) & \text{if } r_j - r'_j + r'_j \leq 0 \end{array} \right.
$$
$$
= \sum_{j=1}^{p} \left\{ \begin{array}{ll} \tau_j(r_j - r'_j) + \tau_j r'_j & \text{if } r_j - r'_j + r'_j \geq 0 \\ (\tau_j - 1)(r_j - r'_j) + (\tau_j - 1)r'_j & \text{if } r_j - r'_j + r'_j \leq 0 \end{array} \right.
$$
$$
\leq \sum_{j=1}^{p} \left\{ \begin{array}{ll} |r_j - r'_j| + \tau_j r'_j & \text{if } r_j - r'_j + r'_j \geq 0 \\ |r_j - r'_j| + (\tau_j - 1)r'_j & \text{if } r_j - r'_j + r'_j \leq 0 \end{array} \right.
$$
$$
\leq \sum_{j=1}^{p} \left( |r_j - r'_j| + \max(\tau_j r'_j, (\tau_j - 1)r'_j) \right)
$$
$$
= \|\boldsymbol{r} - \boldsymbol{r}'\|_{\ell_1} + \ell_\tau(\boldsymbol{r}').
$$

Switching $\boldsymbol{r}$ and $\boldsymbol{r}'$ we get $|\ell_\tau(\boldsymbol{r}) - \ell_\tau(\boldsymbol{r}')| \leq \|\boldsymbol{r} - \boldsymbol{r}'\|_{\ell_1}$. Since by Cauchy-Schwarz inequality $\|\boldsymbol{r} - \boldsymbol{r}'\|_{\ell_1} \leq \sqrt{p}\|\boldsymbol{r} - \boldsymbol{r}'\|_{\ell_2}$, we obtain that $\ell_\tau$ is $\sqrt{p}$-Lipschitz.

In addition, $\ell_\tau$ is bounded for the residuals of interest:

$$
\forall (f, \mathbf{x}, \boldsymbol{y}) \in \mathcal{F} \times \mathcal{X} \times \mathcal{Y}^p,
$$
$$
0 \leq \ell_\tau(\boldsymbol{y} - f(\mathbf{x}) - \boldsymbol{b}) \leq \|\boldsymbol{y} - f(\mathbf{x}) - \boldsymbol{b}\|_{\ell_1}
$$
$$
\leq \|\boldsymbol{y} - \boldsymbol{b}\|_{\ell_1} + \|f(\mathbf{x})\|_{\ell_1}
$$
$$
\leq 2pa + \sqrt{p}\|f(\mathbf{x})\|_{\ell_2}
$$
$$
\leq 2pa + c\sqrt{p}\kappa.
$$

Let $\mathcal{U} = \{u \colon (\mathbf{x}, \boldsymbol{y}) \in \mathcal{X} \times \mathbb{R}^p \mapsto \boldsymbol{y} - f(\mathbf{x}) - \boldsymbol{b}, f \in \mathcal{F}\}$. By Proposition A.1 we have with probability at least $1 - \delta$:

$$
\sup_{u \in \mathcal{U}} \left( \mathbb{E}\left[\ell_\tau(u(X, Y))\right] - \frac{1}{n} \sum_{i=1}^{n} \ell_\tau(u(X_i, Y_i)) \right) \leq 2\sqrt{2p}\mathcal{R}_n(\mathcal{U}) + (2pa + c\sqrt{p}\kappa)\sqrt{\frac{\log(1/\delta)}{2n}}.
$$

Let $(\epsilon_{i,j})_{\substack{1 \leq i \leq n \\ 1 \leq j \leq p}}$ be an *iid* sample of Rademacher random variables. Then:

$$
\begin{aligned}
\mathcal{R}_n(\mathcal{U}) &= \mathbb{E}\left[\sup_{u \in \mathcal{U}}\left(\frac{1}{n}\sum_{\substack{1 \leq i \leq n \\ 1 \leq j \leq p}} \epsilon_{i,j} u_j(X_i, Y_i)\right)\right] \\
&= \mathbb{E}\left[\sup_{f \in \mathcal{F}}\left(\frac{1}{n}\sum_{\substack{1 \leq i \leq n \\ 1 \leq j \leq p}} \epsilon_{i,j}(Y_i - f_j(X_i) - b_j)\right)\right] \\
&= \mathbb{E}\left[\sup_{f \in \mathcal{F}}\left(\frac{1}{n}\sum_{\substack{1 \leq i \leq n \\ 1 \leq j \leq p}} \epsilon_{i,j} f_j(X_i)\right)\right] \\
&\quad + \mathbb{E}\left[\frac{1}{n}\sum_{\substack{1 \leq i \leq n \\ 1 \leq j \leq p}} \epsilon_{i,j} Y_i\right] + b_j\mathbb{E}\left[\frac{1}{n}\sum_{\substack{1 \leq i \leq n \\ 1 \leq j \leq p}} \epsilon_{i,j}\right] \\
&= \mathbb{E}\left[\sup_{f \in \mathcal{F}}\left(\frac{1}{n}\sum_{i=1}^{n} \epsilon_{i,j} f_j(X_i)\right)\right] \\
&\quad + \frac{1}{n}\sum_{\substack{1 \leq i \leq n \\ 1 \leq j \leq p}} \mathbb{E}\left[\epsilon_{i,j}\right]\mathbb{E}\left[Y_i\right] + b_j\frac{1}{n}\sum_{\substack{1 \leq i \leq n \\ 1 \leq j \leq p}} \mathbb{E}\left[\epsilon_{i,j}\right] \\
&= \mathbb{E}\left[\sup_{f \in \mathcal{F}}\left(\frac{1}{n}\sum_{\substack{1 \leq i \leq n \\ 1 \leq j \leq p}} \epsilon_{i,j} f_j(X_i)\right)\right] \\
&= \mathcal{R}_n(\mathcal{F}) \\
&\leq c\sqrt{\frac{\kappa}{n}} \quad\quad\quad\quad\quad\quad\quad\quad\quad \text{(Proposition A.2).}
\end{aligned}
$$

This concludes the proof. $\qquad\qquad\qquad\qquad\qquad\qquad\qquad\qquad\qquad\qquad\qquad$ $\square$

## A.2 Quantile deviation

Given a vector of probabilities $\boldsymbol{\tau} \in (0,1)^p$ and a quantile estimator $\hat{h}\colon \mathcal{X} \to \mathbb{R}^p$, we are interested in controlling the deviation of $\mathbb{E}\left[\mathbb{P}\left(Y \leq \hat{h}_j(X) \mid X\right)\right]$ from $\tau_j$ (for a particular $j \in \mathbb{N}_p$). For this purpose, we would like to derive a uniform bound using the scalar counterpart of Proposition A.1 (which is identical but substituting $\sqrt{2}$ by 1 [3]). Since such a bound is true for all hypothesis $h$, we do not require $\mathbb{E}\left[\mathbb{P}\left(Y \leq h_j(X) \mid X\right)\right]$ to be close to $\tau_j$, but to its empirical twin $\frac{1}{n}\sum_{i=1}^{n} \mathrm{I}_{\mathbb{R}_-}\left(Y_i - h_j(X_i)\right)$, where $\mathrm{I}_{\mathbb{R}_-}$ is the indicator function of the set $\mathbb{R}_-$. For a quantile estimator $\hat{h}$, the quantile property states that $\tau_j$ is sufficiently close to $\frac{1}{n}\sum_{i=1}^{n} \mathrm{I}_{\mathbb{R}_-}\left(y_i - \hat{h}_j(\mathbf{x}_i)\right)$.

Now, remark that:

$$
\mathbb{E}\left[\mathbb{P}\left(Y \leq h_j(X) \mid X\right)\right] = \mathbb{E}\left[\mathbb{E}\left[\mathrm{I}_{\mathbb{R}_-}\left(Y - h_j(X)\right) \mid X\right]\right] = \mathbb{E}\left[\mathrm{I}_{\mathbb{R}_-}\left(Y - h_j(X)\right)\right].
$$

Thus, the two quantities to compare clearly appear as an expected and an empirical costs based on the loss function $\mathrm{I}_{\mathbb{R}_-}$. Unfortunately, that loss function is not Lipschitz continuous. In order to circumvent that pitfall, we introduce an artificial margin $\epsilon > 0$ and two ramp functions $\Gamma_\epsilon^-$ and $\Gamma_\epsilon^+$ (see definition in Theorem A.5). These surrogate mappings are $\frac{1}{\epsilon}$-Lipschitz and respectively lower and upper bound $\mathrm{I}_{\mathbb{R}_-}$.

Finally, Theorem A.5 states that $\mathbb{E}\left[\mathbb{P}\left(Y \leq h_j(X) \mid X\right)\right]$ is uniformly bounded by the *empirical quantile levels* $\frac{1}{n}\sum_{i=1}^{n} \Gamma_\epsilon^-(Y_i - h_j(X_i))$ and $\frac{1}{n}\sum_{i=1}^{n} \Gamma_\epsilon^+(Y_i - h_j(X_i))$ to an additive bias in $O(1/\sqrt{n})$.

**Theorem A.5** (Quantile deviation). *Let $\boldsymbol{\tau} \in (0,1)^p$, $((X_i, Y_i))_{1 \le i \le n} \in (\mathcal{X} \times \mathcal{Y})^n$ be iid random variables (independent from $(X,Y)$) and $\boldsymbol{b} \in \mathbb{R}^p$. Let $\mathcal{H} = \{f + \boldsymbol{b} \colon f \in \mathcal{K}_K / \|f\|_{\mathcal{K}} \le c\}$, for a given $c \in \mathbb{R}_+$, be the class of hypotheses. Moreover, assume that there exists $\kappa \in \mathbb{R}_+$ such that: $\sup_{\mathbf{x} \in \mathcal{X}} \operatorname{tr}(K(\mathbf{x}, \mathbf{x})) \le \kappa$. Let $\epsilon > 0$ be an artificial margin,*

$$\Gamma_\epsilon^+ \colon r \in \mathbb{R} \mapsto \operatorname{proj}_{[0,1]} \left( 1 - \frac{r}{\epsilon} \right) \quad and \quad \Gamma_\epsilon^- \colon r \in \mathbb{R} \mapsto \operatorname{proj}_{[0,1]} \left( -\frac{r}{\epsilon} \right),$$

*two ramp functions, $j \in \mathbb{N}_p$ and $\delta \in (0,1]$. Then with probability at least $1 - \delta$:*

$$\forall h \in \mathcal{H} \colon \frac{1}{n} \sum_{i=1}^n \Gamma_\epsilon^-(Y_i - h_j(X_i)) - \Delta \le \mathbb{E}\left[\mathbb{P}\left(Y \le h_j(X) \mid X\right)\right] \le \frac{1}{n} \sum_{i=1}^n \Gamma_\epsilon^+(Y_i - h_j(X_i)) + \Delta,$$

*where $\Delta = \frac{2c}{\epsilon} \sqrt{\frac{\kappa}{n}} + \sqrt{\frac{\log(2/\delta)}{2n}}$.*

*Proof.* First, let us remind that: $\mathbb{E}\left[\mathbb{P}\left(Y \le h_j(X) \mid X\right)\right] = \mathbb{E}\left[\mathbb{I}_{\mathbb{R}_-}\left(Y - h_j(X)\right)\right]$, and $\forall r \in \mathbb{R}, \Gamma_\epsilon^-(r) \le \mathbb{I}_{\mathbb{R}_-}(r) \le \Gamma_\epsilon^+(r)$. Thus,

$$\mathbb{E}\left[\Gamma_\epsilon^-\left(Y - h_j(X)\right)\right] \le \mathbb{E}\left[\mathbb{P}\left(Y \le h_j(X) \mid X\right)\right] \le \mathbb{E}\left[\Gamma_\epsilon^+\left(Y - h_j(X)\right)\right].$$

Then, remarking that $\Gamma_\epsilon^+$ is $\frac{1}{\epsilon}$-Lipschitz, we obtain (by the same reasoning as for the proof of Theorem A.3 and using [3, Theorem 1]):

$$\forall h \in \mathcal{H} \colon \mathbb{E}\left[\Gamma_\epsilon^+\left(Y - h_j(X)\right)\right] \le \frac{1}{n} \sum_{i=1}^n \Gamma_\epsilon^+(Y_i - h_j(X_i)) + \frac{2c}{\epsilon} \sqrt{\frac{\kappa}{n}} + \sqrt{\frac{\log(1/\delta)}{2n}},$$

with probability at least $1 - \delta$. Respectively, for $-\Gamma_\epsilon^-$, with probability at least $1 - \delta$:

$$\forall h \in \mathcal{H} \colon -\mathbb{E}\left[\Gamma_\epsilon^-\left(Y - h_j(X)\right)\right] \le -\frac{1}{n} \sum_{i=1}^n \Gamma_\epsilon^-(Y_i - h_j(X_i)) + \frac{2c}{\epsilon} \sqrt{\frac{\kappa}{n}} + \sqrt{\frac{\log(1/\delta)}{2n}}.$$

Gathering everything with the union bound concludes the proof. $\qquad\square$

## B  Dual formulation

In this section, we derive a dual problem for learning a joint quantile regressor *with non-crossing constraints*. These last constraints are set thanks to a matrix $\boldsymbol{A}$ defined below. Considering $\boldsymbol{A}$ as the null-matrix gives a dual formulation for the learning problem *without non-crossing constraints*.

Let $C \in \mathbb{R}_+$, $\boldsymbol{\tau} \in (0,1)^p$, such that $\tau_j > \tau_{j+1}$ $(\forall j \in \mathbb{N}_{p-1})$ and the finite difference operator, embodied by the matrix:

$$\boldsymbol{A} = \begin{pmatrix} 1 & -1 & 0 & \dots & 0 \\ 0 & 1 & -1 & \ddots & 0 \\ \vdots & \ddots & \ddots & \ddots & 0 \\ 0 & \dots & 0 & 1 & -1 \end{pmatrix} \in \mathbb{R}^{(p-1) \times p}.$$

The primal problem we are interested in (with the associated dual variables) is:

$$\begin{aligned}
&\underset{\substack{f \in \mathcal{H}, \boldsymbol{b} \in \mathbb{R}^p \\ \boldsymbol{\xi}, \boldsymbol{\xi}^* \in (\mathbb{R}^p)^n,}}{\text{minimize}} \quad \frac{1}{2} \|f\|_{\mathcal{K}}^2 + C \sum_{i=1}^n \langle \boldsymbol{\tau} \mid \boldsymbol{\xi}_i \rangle_{\ell_2} + C \sum_{i=1}^n \langle \mathbb{1} - \boldsymbol{\tau} \mid \boldsymbol{\xi}_i^* \rangle_{\ell_2} \\
&\text{s.t.} \quad \left\{ \begin{array}{ll}
\forall i \in \mathbb{N}_n \colon \boldsymbol{y}_i - f(\mathbf{x}_i) - \boldsymbol{b} = \boldsymbol{\xi}_i - \boldsymbol{\xi}_i^* & : \boldsymbol{\alpha}_i \in \mathbb{R}^p \\
\boldsymbol{\xi}_i \succcurlyeq 0 & : \boldsymbol{\beta}_i \in \mathbb{R}_+^p \\
\boldsymbol{\xi}_i^* \succcurlyeq 0 & : \boldsymbol{\gamma}_i \in \mathbb{R}_+^p \\
\boldsymbol{A}(f(\mathbf{x}_i) + b) \succcurlyeq 0 & : \boldsymbol{\delta}_i \in \mathbb{R}_+^{p-1}.
\end{array} \right.
\end{aligned}$$

The last constraint enforces the regressors not to cross on the training points (hard non-crossing constraints). Let us write the Lagrangian function:

$$\mathfrak{L}(f, \boldsymbol{b}, \boldsymbol{\xi}_i, \boldsymbol{\xi}_i^*, \boldsymbol{\alpha}_i, \boldsymbol{\beta}_i, \boldsymbol{\gamma}_i, \boldsymbol{\delta}_i) = \frac{1}{2} \|f\|_{\mathcal{K}}^2 + C \sum_{i=1}^n \langle \boldsymbol{\tau} \mid \boldsymbol{\xi}_i \rangle_{\ell_2} + C \sum_{i=1}^n \langle \mathbb{1} - \boldsymbol{\tau} \mid \boldsymbol{\xi}_i^* \rangle_{\ell_2}$$

$$+ \sum_{i=1}^n \langle \boldsymbol{\alpha}_i \mid \boldsymbol{y}_i \rangle_{\ell_2} - \sum_{i=1}^n \langle \boldsymbol{\alpha}_i \mid f(\mathbf{x}_i) \rangle_{\ell_2} - \sum_{i=1}^n \langle \boldsymbol{\alpha}_i \mid \boldsymbol{b} \rangle_{\ell_2}$$

$$- \sum_{i=1}^n \langle \boldsymbol{\alpha}_i \mid \boldsymbol{\xi}_i \rangle_{\ell_2} + \sum_{i=1}^n \langle \boldsymbol{\alpha}_i \mid \boldsymbol{\xi}_i^* \rangle_{\ell_2}$$

$$- \sum_{i=1}^n \langle \boldsymbol{\beta}_i \mid \boldsymbol{\xi}_i \rangle_{\ell_2} - \sum_{i=1}^n \langle \boldsymbol{\gamma}_i \mid \boldsymbol{\xi}_i^* \rangle_{\ell_2}$$

$$- \sum_{i=1}^n \langle \boldsymbol{\delta}_i \mid \boldsymbol{A}(f(\mathbf{x}_i) + b) \rangle_{\ell_2}$$

$$= \frac{1}{2} \|f\|_{\mathcal{K}}^2 - \left\langle \sum_{i=1}^n K_{\mathbf{x}_i}(\boldsymbol{\alpha}_i + \boldsymbol{A}^\top \boldsymbol{\delta}_i) \mid f \right\rangle_{\mathcal{H}} + \sum_{i=1}^n \langle C\boldsymbol{\tau} - \boldsymbol{\alpha}_i - \boldsymbol{\beta}_i \mid \boldsymbol{\xi}_i \rangle_{\ell_2}$$

$$+ \sum_{i=1}^n \langle C(\mathbb{1} - \boldsymbol{\tau}) + \boldsymbol{\alpha}_i - \boldsymbol{\gamma}_i \mid \boldsymbol{\xi}_i^* \rangle_{\ell_2} - \left\langle \sum_{i=1}^n (\boldsymbol{\alpha}_i + \boldsymbol{A}^\top \boldsymbol{\delta}_i) \mid \boldsymbol{b} \right\rangle_{\ell_2}$$

$$+ \sum_{i=1}^n \langle \boldsymbol{\alpha}_i \mid \boldsymbol{y}_i \rangle_{\ell_2}.$$

First order optimality conditions for the primal variables give:

$$\begin{cases} \nabla_f \mathfrak{L}(f, \boldsymbol{b}, \boldsymbol{\xi}_i, \boldsymbol{\xi}_i^*, \boldsymbol{\alpha}_i, \boldsymbol{\beta}_i, \boldsymbol{\gamma}_i, \boldsymbol{\delta}_i) = f - \sum_{i=1}^n K_{\mathbf{x}_i}(\boldsymbol{\alpha}_i + \boldsymbol{A}^\top \boldsymbol{\delta}_i) = 0 \\ \nabla_b \mathfrak{L}(f, \boldsymbol{b}, \boldsymbol{\xi}_i, \boldsymbol{\xi}_i^*, \boldsymbol{\alpha}_i, \boldsymbol{\beta}_i, \boldsymbol{\gamma}_i, \boldsymbol{\delta}_i) = - \sum_{i=1}^n (\boldsymbol{\alpha}_i + \boldsymbol{A}^\top \boldsymbol{\delta}_i) = 0 \\ \nabla_{\boldsymbol{\xi}_i} \mathfrak{L}(f, \boldsymbol{b}, \boldsymbol{\xi}_i, \boldsymbol{\xi}_i^*, \boldsymbol{\alpha}_i, \boldsymbol{\beta}_i, \boldsymbol{\gamma}_i, \boldsymbol{\delta}_i) = C\boldsymbol{\tau} - \boldsymbol{\alpha}_i - \boldsymbol{\beta}_i = 0 \\ \nabla_{\boldsymbol{\xi}_i^*} \mathfrak{L}(f, \boldsymbol{b}, \boldsymbol{\xi}_i, \boldsymbol{\xi}_i^*, \boldsymbol{\alpha}_i, \boldsymbol{\beta}_i, \boldsymbol{\gamma}_i, \boldsymbol{\delta}_i) = C(\mathbb{1} - \boldsymbol{\tau}) + \boldsymbol{\alpha}_i - \boldsymbol{\gamma}_i = 0. \end{cases}$$

That is:

$$\begin{cases} f = \sum_{i=1}^n K_{\mathbf{x}_i}(\boldsymbol{\alpha}_i + \boldsymbol{A}^\top \boldsymbol{\delta}_i) \\ 0 = \sum_{i=1}^n (\boldsymbol{\alpha}_i + \boldsymbol{A}^\top \boldsymbol{\delta}_i) \\ \boldsymbol{\beta}_i = C\boldsymbol{\tau} - \boldsymbol{\alpha}_i \\ \boldsymbol{\gamma}_i = C(\mathbb{1} - \boldsymbol{\tau}) + \boldsymbol{\alpha}_i. \end{cases}$$

Recalling that $\boldsymbol{\beta}_i \succcurlyeq 0$ and $\boldsymbol{\gamma}_i \succcurlyeq 0$, we obtain: $C(\boldsymbol{\tau} - \mathbb{1}) \preccurlyeq \boldsymbol{\alpha}_i \preccurlyeq C\boldsymbol{\tau}$. Then, by substitution of the first order equations in the Lagrangian function, the linear expressions in the primal variables vanish and the quadratic part becomes:

$$\frac{1}{2} \|f\|_{\mathcal{K}}^2 - \left\langle \sum_{i=1}^n K_{\mathbf{x}_i}(\boldsymbol{\alpha}_i + \boldsymbol{A}^\top \boldsymbol{\delta}_i) \mid f \right\rangle_{\mathcal{H}} = \frac{1}{2} \sum_{i,j=1}^n \left\langle K_{\mathbf{x}_i}(\boldsymbol{\alpha}_i + \boldsymbol{A}^\top \boldsymbol{\delta}_i) \mid K_{\mathbf{x}_j}(\boldsymbol{\alpha}_j + \boldsymbol{A}^\top \boldsymbol{\delta}_j) \right\rangle_{\mathcal{H}}$$

$$- \sum_{i,j=1}^n \left\langle K_{\mathbf{x}_i}(\boldsymbol{\alpha}_i + \boldsymbol{A}^\top \boldsymbol{\delta}_i) \mid K_{\mathbf{x}_j}(\boldsymbol{\alpha}_j + \boldsymbol{A}^\top \boldsymbol{\delta}_j) \right\rangle_{\mathcal{H}}$$

$$= -\frac{1}{2} \sum_{i,j=1}^n \left\langle (\boldsymbol{\alpha}_i + \boldsymbol{A}^\top \boldsymbol{\delta}_i) \mid K(\mathbf{x}_i, \mathbf{x}_j)(\boldsymbol{\alpha}_j + \boldsymbol{A}^\top \boldsymbol{\delta}_j) \right\rangle_{\ell_2}.$$

Gathering every thing, the dual problem writes:

$$\underset{\substack{\boldsymbol{\alpha}_i \in \mathbb{R}^P, \boldsymbol{\delta}_i \in \mathbb{R}^{P-1} \\ \forall i \in \mathbb{N}_n}}{\text{maximize}} \quad -\frac{1}{2}\sum_{i,j=1}^n \left\langle (\boldsymbol{\alpha}_i + \boldsymbol{A}^\top \boldsymbol{\delta}_i) \mid K(\mathbf{x}_i, \mathbf{x}_j)(\boldsymbol{\alpha}_j + \boldsymbol{A}^\top \boldsymbol{\delta}_j) \right\rangle_{\ell_2} + \sum_{i=1}^n \left\langle \boldsymbol{\alpha}_i \mid \boldsymbol{y}_i \right\rangle_{\ell_2}$$

$$\text{s.\,t.} \quad \begin{cases} \forall i \in \mathbb{N}_n : C(\boldsymbol{\tau} - \mathbb{1}) \preccurlyeq \boldsymbol{\alpha}_i \preccurlyeq C\boldsymbol{\tau} \\ \qquad\qquad \boldsymbol{\delta}_i \succcurlyeq 0 \\ \sum_{i=1}^n (\boldsymbol{\alpha}_i + \boldsymbol{A}^\top \boldsymbol{\delta}_i) = 0_{\mathbb{R}^p}. \end{cases}$$

In order to simplify the previous problem, let $\boldsymbol{u}_i = \boldsymbol{\alpha}_i + \boldsymbol{A}^\top \boldsymbol{\delta}_i$ and remark that $\boldsymbol{A}\boldsymbol{y}_i = y_i(\boldsymbol{A}\mathbb{1}) = 0$. The new dual problem then becomes:

$$\underset{\substack{\boldsymbol{u}_i \in \mathbb{R}^P, \boldsymbol{\delta}_i \in \mathbb{R}^{P-1} \\ \forall i \in \mathbb{N}_n}}{\text{maximize}} \quad -\frac{1}{2}\sum_{i,j=1}^n \left\langle \boldsymbol{u}_i \mid K(\mathbf{x}_i, \mathbf{x}_j)\boldsymbol{u}_j \right\rangle_{\ell_2} + \sum_{i=1}^n \left\langle \boldsymbol{u}_i \mid \boldsymbol{y}_i \right\rangle_{\ell_2}$$

$$\text{s.\,t.} \quad \begin{cases} \forall i \in \mathbb{N}_n : C(\boldsymbol{\tau} - \mathbb{1}) \preccurlyeq \boldsymbol{u}_i - \boldsymbol{A}^\top \boldsymbol{\delta}_i \preccurlyeq C\boldsymbol{\tau} \\ \qquad\qquad \boldsymbol{\delta}_i \succcurlyeq 0 \\ \sum_{i=1}^n \boldsymbol{u}_i = 0_{\mathbb{R}^p}. \end{cases}$$

Primal variables are recovered thanks to first order conditions. First, $f = \sum_{i=1}^n K_{\mathbf{x}_i} \boldsymbol{u}_i$. Second, the intercept $\boldsymbol{b}$ can be obtained either by detecting couples $(i, \ell) \in \mathbb{N}_n \times \mathbb{N}_p$ such that $C(\tau_\ell - 1) < (u_i)_\ell - (\boldsymbol{A}^\top \boldsymbol{\delta}_i)_\ell < C\tau_\ell$ (in this case $b_\ell = y_i - f_\ell(\mathbf{x}_i)$), or by remarking that $\boldsymbol{b}$ is a dual vector for the linear constraint $\sum_{i=1}^n \boldsymbol{u}_i = 0_{\mathbb{R}^p}$ (if one uses a primal-dual algorithm to solve the previous optimization problem).

When non-crossing constraints are dismissed ($\boldsymbol{A} = 0_{\mathbb{R}^{(p-1) \times p}}$), the regressor $h = f + \boldsymbol{b}$ satisfies the quantile property. Thus, knowing $f$, the intercepts $b_\ell$ can be recovered as $\tau_\ell$-quantiles of $(y_i - f_\ell(\mathbf{x}_i))_{1 \le i \le n}$.

## C  Algorithmic details

This section details an augmented Lagrangian scheme for estimating quantile regressors. We start with the dual formulation of the learning problem (without non-crossing constraints):

$$\underset{\boldsymbol{\alpha} \in (\mathbb{R}^p)^n}{\text{minimize}} \quad \frac{1}{2}\sum_{i,j=1}^n \left\langle \boldsymbol{\alpha}_i \mid K(\mathbf{x}_i, \mathbf{x}_j)\boldsymbol{\alpha}_j \right\rangle_{\ell_2} - \sum_{i=1}^n y_i \left\langle \boldsymbol{\alpha}_i \mid \mathbb{1} \right\rangle_{\ell_2}$$

$$\text{s.\,t.} \quad \begin{cases} \forall i \in \mathbb{N}_n : C(\boldsymbol{\tau} - \mathbb{1}) \preccurlyeq \boldsymbol{\alpha}_i \preccurlyeq C\boldsymbol{\tau} \\ \sum_{i=1}^n \boldsymbol{\alpha}_i = 0_{\mathbb{R}^p}. \end{cases}$$

The method consists in solving the saddle point problem with an additional squared penalty [2]:

$$\underset{\boldsymbol{b} \in \mathbb{R}^p}{\text{maximize}} \ \underset{\boldsymbol{\alpha} \in (\mathbb{R}^p)^n}{\text{minimize}} \quad \frac{1}{2}\sum_{i,j=1}^n \left\langle \boldsymbol{\alpha}_i \mid K(\mathbf{x}_i, \mathbf{x}_j)\boldsymbol{\alpha}_j \right\rangle_{\ell_2} - \sum_{i=1}^n y_i \left\langle \boldsymbol{\alpha}_i \mid \mathbb{1} \right\rangle_{\ell_2}$$

$$+ \left\langle \boldsymbol{b} \mid \sum_{i=1}^n \boldsymbol{\alpha}_i \right\rangle_{\ell_2} + \frac{\mu}{2}\left\| \sum_{i=1}^n \boldsymbol{\alpha}_i \right\|_{\ell_2}^2$$

$$\text{s.\,t.} \quad \forall i \in \mathbb{N}_n : C(\boldsymbol{\tau} - \mathbb{1}) \preccurlyeq \boldsymbol{\alpha}_i \preccurlyeq C\boldsymbol{\tau},$$

where $\mu$ is a positive scalar. The next step is to split the optimization program into an outer problem (depending only on the variable $\boldsymbol{b}$) and an inner one (depending on $\boldsymbol{\alpha}$). This latter problem is:

$$\underset{\boldsymbol{\alpha} \in (\mathbb{R}^p)^n}{\text{minimize}} \quad \frac{1}{2}\sum_{i,j=1}^n \left\langle \boldsymbol{\alpha}_i \mid (K(\mathbf{x}_i, \mathbf{x}_j) + \mu\boldsymbol{I})\boldsymbol{\alpha}_j \right\rangle_{\ell_2} + \sum_{i=1}^n \left\langle \boldsymbol{\alpha}_i \mid \boldsymbol{b} - y_i\mathbb{1} \right\rangle_{\ell_2} \qquad (1)$$

$$\text{s.\,t.} \quad \forall i \in \mathbb{N}_n : C(\boldsymbol{\tau} - \mathbb{1}) \preccurlyeq \boldsymbol{\alpha}_i \preccurlyeq C\boldsymbol{\tau},$$

**Algorithm 1** Augmented Lagrangian algorithm

---

Initialize $\mu \leftarrow 10$, $\boldsymbol{b} \leftarrow 0_{\mathbb{R}^p}$.
**repeat**
    Solve Optimization Problem (1).
    Make a gradient step on $\boldsymbol{b}$ with step size $\mu$.
**until** $\left\| \sum_{i=1}^{n} \boldsymbol{\alpha}_i \right\|_{\ell_2}^2$ is small enough

---

Table 1: Empirical quantile loss $\times 100$ (the closer to 0, the better).

| Data set | IND. | IND. (NC) | MTFL | JQR |
|---|---|---|---|---|
| caution | $4.50 \pm 39.08$ | $\mathbf{3.33} \pm 37.84$ | $7.00 \pm 32.57$ | ∘∘∘ $7.17 \pm 36.40$ |
| ftcollinssnow | $1.43 \pm 38.49$ | $1.79 \pm 38.12$ | $1.25 \pm 38.50$ | ∘∘∘ $\mathbf{0.54} \pm 35.77$ |
| highway | $10.83 \pm 70.12$ | $10.83 \pm 71.20$ | $\mathbf{7.50} \pm 62.96$ | ∘∘∘ $15.00 \pm 67.11$ |
| heights | $-1.15 \pm 9.88$ | $-1.14 \pm 9.88$ | $\mathbf{-0.76} \pm 9.39$ | ∘∘∘ $-1.26 \pm 9.31$ |
| sniffer | $-6.58 \pm 26.45$ | $-6.58 \pm 27.59$ | $\mathbf{-3.95} \pm 27.57$ | ∘∘∘ $-4.08 \pm 29.23$ |
| snowgeese | $\mathbf{-0.00} \pm 44.03$ | $1.07 \pm 43.41$ | $1.43 \pm 50.03$ | ∘∘∘ $-6.43 \pm 44.94$ |
| ufc | $0.58 \pm 12.13$ | $0.89 \pm 11.97$ | $\mathbf{-0.31} \pm 13.59$ | ●●∘ $-1.79 \pm 13.42$ |
| birthwt | $2.02 \pm 29.80$ | $2.02 \pm 29.80$ | $2.11 \pm 34.55$ | ∘∘∘ $\mathbf{0.88} \pm 33.86$ |
| crabs | $-2.42 \pm 20.52$ | $-1.25 \pm 22.08$ | $-2.17 \pm 22.21$ | ∘∘∘ $\mathbf{-0.50} \pm 21.75$ |
| GAGurine | $1.95 \pm 17.43$ | $1.74 \pm 17.39$ | $\mathbf{0.89} \pm 16.71$ | ∘∘∘ $1.84 \pm 16.89$ |
| geyser | $1.22 \pm 18.84$ | $1.17 \pm 18.66$ | $\mathbf{0.22} \pm 19.20$ | ∘∘∘ $1.61 \pm 19.04$ |
| gilgais | $0.95 \pm 18.35$ | $0.95 \pm 18.19$ | $-0.64 \pm 16.12$ | ∘∘∘ $\mathbf{0.18} \pm 20.22$ |
| topo | $\mathbf{-19.38} \pm 70.18$ | $-19.38 \pm 71.34$ | $-20.00 \pm 64.18$ | ∘∘∘ $-20.31 \pm 65.73$ |
| BostonHousing | $7.40 \pm 18.26$ | $7.30 \pm 17.94$ | $5.72 \pm 16.72$ | ∘∘∘ $\mathbf{5.30} \pm 15.74$ |
| CobarOre | $26.67 \pm 72.49$ | $25.42 \pm 72.58$ | $-22.08 \pm 77.16$ | ∘∘∘ $\mathbf{20.00} \pm 72.62$ |
| engel | $-3.66 \pm 18.71$ | $-3.73 \pm 18.72$ | $-3.73 \pm 19.73$ | ∘∘∘ $\mathbf{-2.39} \pm 17.18$ |
| mcycle | $\mathbf{1.37} \pm 29.20$ | $1.75 \pm 30.73$ | $4.75 \pm 28.42$ | ∘∘∘ $6.25 \pm 30.84$ |
| BigMac2003 | $-4.76 \pm 51.10$ | $\mathbf{-0.00} \pm 45.24$ | $1.67 \pm 52.19$ | ∘∘∘ $0.24 \pm 45.76$ |
| UN3 | $4.44 \pm 19.11$ | $4.21 \pm 19.23$ | $\mathbf{2.62} \pm 17.06$ | ∘∘∘ $4.76 \pm 20.68$ |
| cpus | $2.38 \pm 20.30$ | $3.57 \pm 19.63$ | $\mathbf{1.51} \pm 15.02$ | ∘∘∘ $1.67 \pm 31.78$ |

where $\boldsymbol{I}$ is the identity matrix. This inner optimization problem is a quadratic program with a box constraint. This is quite easily solvable. Thus, following [2], we can learn quantile estimators thanks to the simple alternate scheme described in Algorithm 1. In practice, the inner solver used in Algorithm 1 in order to get an approximate solution for Problem (1) is the primal dual-dual coordinate descent proposed in [6] with $10^4$ as the maximum number of iterations.

# D    Numerical results

## D.1    Quantile regression

Another criterion for assessing quantile regression methods is the quantile loss $\sum_{j=1}^{p} \left[ \left[ \frac{1}{n} \sum_{i=1}^{n} \mathrm{I}_{\mathbb{R}_-} (y_i - h_j(\mathbf{x}_i)) \right] - \tau_j \right]$, where $\mathrm{I}_{\mathbb{R}_-}$ is the indicator function of the set $\mathbb{R}_-$. This loss measures the deviation of the estimators $h_j$ to the prescribed quantile levels $\tau_j$.

However, the quantile loss is quite an equivocal criterion, since it measures *how much* the *unconditional* quantile property is satisfied. This unconditional indicator is indeed the only way to get a piece of information concerning the *conditional* quantile property. For instance, Takeuchi et al. [8] empirically showed (with the same datasets) that the constant function based on the unconditional quantile estimator performs best under this criterion, even though it is expected to be a poor conditional quantile regressor. The numerical results in Table 1 follow this remark and the results previously obtained [8]. No significant ranking comes out.

## D.2    Training algorithms

In order to compare the implementations of the three algorithms for solving the dual optimization problem of joint quantile regression, the following procedure has been set up: we first run QP, with a relative tolerance set to $10^{-2}$, and store the optimal objective value. Then, the two other methods (AUG. LAG and PDCD) are launched and stopped when they pass the objective value reached by QP.

During the descent of PDCD, we used an efficient accumulated objective value, which is not exact since the iterate $\boldsymbol{\alpha}$ is not feasible to the linear constraint. Table 2 describes the average objective values (divided by the sample size) reached by each algorithm after projection of the best candidate

Table 2: Average objective value (divided by the sample size) reached in the second numerical experiment presented in the corpus of the paper.

| Size | QP | Aug. Lag. | PDCD |
|------|-----|-----------|------|
| 250 | -109.69 ± 5.41 | -109.73 ± 5.41 | **-109.72** ± 5.40 |
| 500 | -109.84 ± 2.09 | -109.85 ± 2.08 | **-109.88** ± 2.11 |
| 1000 | -104.13 ± 1.49 | – | **-104.17** ± 1.49 |
| 2000 | -106.35 ± 2.36 | – | **-106.39** ± 2.38 |

onto the set of constraints. We can check that our approach (PDCD) always reaches a smaller objective value than the target QP. This validates our procedure.