[Reviews · NeurIPS 2016]

Reviewer 1

Summary

The authors introduce a new approach towards quantile regression which doesn't suffer from limitations such as quantile crossing or complicated learning strategies and post-processing. The experiments show improved predictive performance and runtimes.

Qualitative Assessment

Pro: - very well-written paper - clear motivations - strong experimental results Contra: - assumes a lot of advanced knowledge on quantile regression Major comments: Although in general the paper is very well written and most of the steps are adequately motivated, it is hard to read for someone not familiar with the domain. For example, it would make the paper more accessible to a wider audience if some concepts such as expectiles, quantiles, compact set, ramp functions, ... would be formally defined. Moreover, in Sections 3 and 4 I miss the intuition behind all the deductions. In my opinion, solving these issues would make the paper (of which the methodology seems correct and the experimental results convincing) a lot stronger. Minor comments: You perform a Wilcoxon test between JQR and every other method for every dataset. Normally, I would expect that each dataset forms one independent sample, and you count wins/losses on all datasets. How do you define the wins and losses here? Small spelling errors in Section 4 and 6: - line 161: an hypothesis -> a hypothesis - line 174: does not deviate to much -> too much - line 248: a honorable comparison -> an honorable

Confidence in this Review

1-Less confident (might not have understood significant parts)


Reviewer 2

Summary

The paper describes a method to estimate the quantiles of a conditional distribution. Typically multiple conditional quantile functions are needed to give a full statistical description of a particular dataset. Estimation of several conditional quantile functions can cause two or more estimated functions to overlap. This crossing behaviour, however, should not happen as the true quantile functions are defined to be non-crossing. To address the need for non-crossing quantile functions, this manuscript resorts to a joint estimation of multiple conditional quantile functions under a multi-task learning framework (as in prior work of [27]). The novelty lies in the usage of kernel methods for vector output (the vector output here corresponds to several quantiles). A simple separable kernel, a product of input kernel and output kernel, is utilised. Experiments are performed on multiple UCI and R datasets, as well as, census and health statistics data on the pinball loss and crossing loss metric.

Qualitative Assessment

Technical quality: 1) Experiments are performed on several datasets and performance metrics are ranging from pinball loss (for accurateness of quantile estimation), non-crossing loss (for compliance w.r.t. the true non-crossing property between quantiles) and running time (though running time of proposed method is not compared w.r.t. baseline methods). However, the 20 datasets used are based on earlier work in 2006 [26] with maximum sample size of only 1375. 2) Theorem 4.1. is appreciated but is restricted as joint estimation of quantile functions involves infinitely many tasks parameterized by a continuous parameter (line 169-171). 3) Re. non-crossing constraint, note that the method of [26] can impose the non-crossing constraint at each of the test points (section 5.2.1. of [26]) to be evaluated (c.f. line 269-270 "the crossing loss is null on the training data but is not guaranteed to be null on the test data"). Novelty/originality: 1) Novelty is rather low. The formulation of multi-task learning for multiple conditional quantile functions was proposed in [27]. The current manuscript encourages the usage of kernel methods for vector-valued functions in which the output kernel encodes the non-crossing condition. Potential impact or usefulness: 1) Baseline methods are cross-validated w.r.t. regularisation parameter C hyper-param only, while the proposed method is cross-validated w.r.t. C and output kernel width gamma. The kernel width of the input space is set to 0.7 quantile of pairwise distances, this should be cross-validated as well especially so for the baseline methods. 2) With the current scale of the experiments (maximum size of datasets of 2000), the impact would be rather low. In fact future work points to the usage of random Fourier features for scalability, when coupled with primal-dual coordinate descent will strengthen the current manuscript.

Confidence in this Review

2-Confident (read it all; understood it all reasonably well)


Reviewer 3

Summary

The authors consider the problem of quantile regression. That is, given pairs (X,Y), the problem is to estimate specified quantiles of the conditional distributions Y|X=x in contrast to 'normal' regression in which the expectation E(Y|X=x) is to be estimated. Previous papers have phrased this as a regression problem for which functions from an RKHS with norm less than some constant are the hypothesis set. The main contribution of this paper is to propose a method to tackle the so-called 'crossing problem', an issue that arises when curves for multiple quantiles are estimated simultaneously and the resulting curves cross over. The method the authors propose is to consider the problem as regression in a vector-valued RKHS. The kernel of the RKHS can be chosen in order to penalise the coordinate-wise functions crossing over. They prove two theoretical results concerning the empirical risk minimiser of their problem. First, that the generalisation error is small with high probability; second, that the error in estimating the quantiles is small with high probability. They write down the problem as an optimisation problem, and solve this using Primal-Dual Coordinate Descent. Finally, they test their method on real datasets, showing that their method performs favorably compared to previous methods.

Qualitative Assessment

Novelty/Originality: The idea to apply vector-valued RKHSs to the problem of quantile regression appears to be novel. I hadn't encountered vector-valued RKHSs before reading this, but I don't know that much about this area. In any case, I like the idea because it is simple but appears to be effective. Although the authors restrict themselves to considering decomposable kernels, I would be interested to hear some discussion of other matrix-valued kernels and the resulting regression functions they would induce. Technical Quality: The authors have provided two theoretical results about their method. I would appreciate more explanation of Theorem 4.2 (quantile property) as the statement of the theorem is a bit difficult to parse. I'd also appreciate some more discussion on selection of the parameter gamma - it seems like it should be interpretable in some way. At the end of section 6.1, it is stated that the pinball loss embraces the crossing loss as a subcriterion. I can see intuitively that this makes sense, but I'm confused as to why the crossing loss would then sometimes be so bad with the previous methods, even though they were also cross-validated using the pinball loss. The following view might be unfashionable, but I would like to see the method being tested on synthetic data for which we know the conditional distributions Y|X=x and so can compare the estimated quantiles to the true quantiles. in addition to the real data experiments. Potential impact: While I am not aware of any particular uses of quantile regression, it seems like it ought to be something that is practically used in the application areas listen in the introduction. Clarity and presentation: In general, I thought that the section layout of the paper was good and the logic was clear. However, the style of writing was at times quite casual and there were a few grammatical errors.

Confidence in this Review

1-Less confident (might not have understood significant parts)


Reviewer 4

Summary

The paper deals with a multiple quantile estimation problem using vector-valued RKHS. Although estimating conditional \tau-quantile function can be achieved by minimizing pinball loss, the so-called crossing problem comes up when optimizing multiple \tau_j quantile functions independently. To tackle this problem, using the theory of vector-valued RKHS with an appropriate matrix-valued kernel(e.g. decomposable kernel) is proposed. Here, the kernel parameter \gamma controls between parallel (and thus non-crossing) curves (when \gamma = 0) and independently estimated curves (when \gamma -> \infty). In the theoretical analysis, they provide uniform generalization bound and bound on the quantile property of the vector-valued RKHS approach. Also, they propose to use Primal-Dual Coordinate Descent algorithm to solve the given optimization problems (1) and (2). Numerical experiments support that their approach shows better performance in terms of both pinball loss and crossing loss.

Qualitative Assessment

This paper nicely shows nonparametric joint quantile estimation algorithm based on vector-valued RKHSs. Applying vector-valued RKHSs to multiple outputs regression problem itself is not really that original; however, the paper looks make enough contribution with theoretical analysis and convincing experimental results. The paper is well-presented in every aspect.

Confidence in this Review

1-Less confident (might not have understood significant parts)


Reviewer 5

Summary

The authors tackle the problem of estimating several quantiles of a regression. Motivated by the embarrassing situation where quantiles cross, the authors seek to overcome this problem by working in a vector valued RKHS that simultaneously estimates each quantiles values, allowing it to preserve the structure in the output space. To solve the novel optimisation problem they make use of tools taken from multi-task learning, allowing an efficient learning algorithm, and is also used for their theoretical analysis of the learned function.

Qualitative Assessment

-> Paper provides an elegant and intuitive solution to an important problem. -> In experiments, might be worthwhile to compare with Gaussian process regressions, which have implicit quantiles.

Confidence in this Review

2-Confident (read it all; understood it all reasonably well)